# REGRESSION WITH LABEL DIFFERENTIAL PRIVACY [*]

**Badih Ghazi**  **Pritish Kamath**  **Ravi Kumar**  **Ethan Leeman**

**Pasin Manurangsi**  **Avinash Varadarajan**  **Chiyuan Zhang**

Google

## ABSTRACT

We study the task of training regression models with the guarantee of *label* differential privacy (DP). Based on a global prior distribution on label values, which could be obtained privately, we derive a label DP randomization mechanism that is optimal under a given regression loss function. We prove that the optimal mechanism takes the form of a "randomized response on bins", and propose an efficient algorithm for finding the optimal bin values. We carry out a thorough experimental evaluation on several datasets demonstrating the efficacy of our algorithm.

## 1 INTRODUCTION

In recent years, differential privacy (DP, Dwork et al., 2006a;b) has emerged as a popular notion of user privacy in machine learning (ML). On a high level, it guarantees that the output model weights remain statistically indistinguishable when any single training example is arbitrarily modified. Numerous DP training algorithms have been proposed, with open-source libraries tightly integrated in popular ML frameworks such as TensorFlow Privacy (Radebaugh & Erlingsson, 2019) and PyTorch Opacus (Yousefpour et al., 2021).

In the context of supervised ML, a training example consists of input features and a target label. While many existing research works focus on protecting both features and labels (e.g., Abadi et al. (2016)), there are also some important scenarios where the input features are already known to the adversary, and thus protecting the privacy of the features is not needed. A canonical example arises from computational advertising where the features are known to one website (a publisher), whereas the conversion events, i.e., the labels, are known to another website (the advertiser).[1] Thus, from the first website's perspective, only the labels can be treated as unknown and private. This motivates the study of *label DP* algorithms, where the statistical indistinguishability is required only when the label of a single example is modified.[2] The study of this model goes back at least to the work of Chaudhuri & Hsu (2011). Recently, several works including (Ghazi et al., 2021a; Malek Esmaeili et al., 2021) studied label DP deep learning algorithms for classification objectives.

**Our Contributions.** In this work, we study label DP for *regression* tasks. We provide a new algorithm that, given a global prior distribution (which, if unknown, could be estimated privately), derives a label DP mechanism that is optimal under a given objective loss function. We provide an explicit characterization of the optimal mechanism for a broad family of objective functions including the most commonly used regression losses such as the Poisson log loss, the mean square error, and the mean absolute error.

More specifically, we show that the optimal mechanism belongs to a class of *randomized response on bins* (Algorithm 1). We show this by writing the optimization problem as a linear program (LP), and characterizing its optimum. With this characterization in mind, it suffices for us to compute the

---

[*]Authors in alphabetical order. Email: {`badih.ghazi`, `ravi.k53`}`@gmail.com`, {`pritishk`, `ethanleeman`, `pasin`, `avaradar`, `chiyuan`}`@google.com`

[1]A similar use case is in mobile advertising, where websites are replaced by apps.

[2]We note that this label DP setting is particularly timely and relevant for ad attribution and conversion measurement given the deprecation of third-party cookies by several browsers and platforms (Wilander, 2020; Wood, 2019; Schuh, 2020).

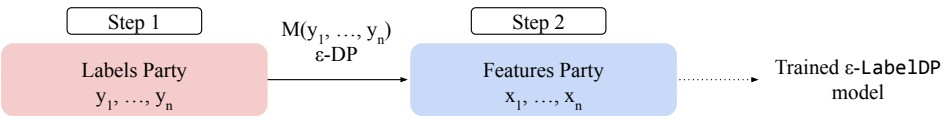

Figure 1: Learning with feature-oblivious label DP.

optimal mechanism among the class of randomized response on bins. We then provide an efficient algorithm for this task, based on dynamic programming (Algorithm 2).

In practice, a prior distribution on the labels is not always available. This leads to our two-step algorithm (Algorithm 3) where we first use a portion of the privacy budget to build an approximate histogram of the labels, and then feed this approximate histogram as a prior into the optimization algorithm in the second step; this step would use the remaining privacy budget. We show that as the number of samples grows, this two-step algorithm yields an expected loss (between the privatized label and the raw label) which is arbitrarily close to the expected loss of the optimal local DP mechanism. (We give a quantitative bound on the convergence rate.)

Our two-step algorithm can be naturally deployed in the two-party learning setting where each example is vertically partitioned with one party holding the features and the other party holding the (sensitive) labels. The algorithm is in fact one-way, requiring a single message to be communicated from the labels party to the features party, and we require that this one-way communication satisfies (label) DP. We refer to this setting, which is depicted in Figure 1, as *feature-oblivious label DP*.

We evaluate our algorithm on three datasets: the 1940 US Census IPUMS dataset, the Criteo Sponsored Search Conversion dataset, and a proprietary app install ads dataset from a commercial mobile app store. We compare our algorithm to several baselines, and demonstrate that it achieves higher utility across all test privacy budgets, with significantly lower test errors for the high privacy regime. For example, for privacy budget $\varepsilon = 0.5$, comparing to the best baseline methods, the test MSE for our algorithm is $\sim 1.5\times$ smaller on the Criteo and US Census datasets, and the relative test error is $\sim 5\times$ smaller on the app ads dataset.

**Organization.** In Section 2, we recall some basics of DP and learning theory, and define the feature-oblivious label DP setting in which our algorithm can be implemented. Our label DP algorithm for regression objectives is presented in Section 3. Our experimental evaluation and results are described in Section 4. A brief overview of related work appears in Section 5. We conclude with some interesting future directions in Section 6. Most proofs are deferred to the Appendix (along with additional experimental details and background material).

## 2 PRELIMINARIES

We consider the standard setting of supervised learning, where we have a set of examples of the form $(x, y) \in \mathcal{X} \times \mathcal{Y}$, drawn from some unknown distribution $\mathcal{D}$ and we wish to learn a predictor $f_\theta$ (parameterized by $\theta$) to minimize $\mathcal{L}(f_\theta) := \mathbb{E}_{(x,y) \sim \mathcal{D}} \ell(f_\theta(x), y)$, for some loss function $\ell : \mathbb{R} \times \mathcal{Y} \to \mathbb{R}_{\geq 0}$; we will consider the case where $\mathcal{Y} \subseteq \mathbb{R}$. Some common loss functions include the *zero-one loss* $\ell_{0\text{-}1}(\tilde{y}, y) := \mathbb{1}[\tilde{y} \neq y]$, the *logistic loss* $\ell_{\log}(\tilde{y}, y) := \log(1 + e^{-\tilde{y}y})$ for binary classification, and the *squared loss* $\ell_{\text{sq}}(\tilde{y}, y) := \frac{1}{2}(\tilde{y} - y)^2$, the *absolute-value loss* $\ell_{\text{abs}}(\tilde{y}, y) := |\tilde{y} - y|$, the *Poisson log loss* $\ell_{\text{Poi}}(\tilde{y}, y) := \tilde{y} - y \cdot \log(\tilde{y})$ for regression. This paper focuses on the regression setting.

However, we wish to perform this learning with differential privacy (DP). We start by recalling the definition of DP, which can be applied to any notion of *adjacent* pairs of datasets. For an overview of DP, we refer the reader to the book of Dwork & Roth (2014).

**Definition 1** (DP; Dwork et al. (2006b))**.** Let $\varepsilon$ be a positive real number. A randomized algorithm $\mathcal{A}$ taking as input a dataset is said to be $\varepsilon$-*differentially private* (denoted $\varepsilon$-DP) if for any two adjacent datasets $X$ and $X'$, and any subset $S$ of outputs of $\mathcal{A}$, we have $\Pr[\mathcal{A}(X) \in S] \leq e^\varepsilon \cdot \Pr[\mathcal{A}(X') \in S]$.

In supervised learning applications, the input to a DP algorithm is the training dataset (i.e., a set of labeled examples) and the output is the description of a predictor (e.g., the weights of the trained

model). Typically, two datasets are considered adjacent if they differ on a single training example; this notion protects both the features and the label of any single example. As discussed in Section 1, in certain situations, protecting the features is either unnecessary or impossible, which motivates the study of label DP that we define next.

**Definition 2** (Label DP; Chaudhuri & Hsu (2011)). An algorithm $\mathcal{A}$ taking as input a training dataset is said to be $\varepsilon$-*label differentially private* (denoted as $\varepsilon$-LabelDP) if it is $\varepsilon$-DP when two input datasets are considered adjacent if they differ on the label of a single training example.

We next define an important special case of training with label DP, which commonly arises in practice. The setting could be viewed as a special case of *vertical federated learning*, where each training example is divided across different parties at the beginning of the training process; see, e.g., the survey of Kairouz et al. (2021) and the references therein. We consider the special case where we only allow a single round of communication from the first party who holds all the labels, to the second party, who holds all the features, and who then trains the ML model and outputs it. We refer to this setting (depicted in Figure 1) as *feature-oblivious label DP* since the label DP property is guaranteed without having to look at the features (but with the ability to look at the labels from all users); we next give the formal definition.

**Definition 3** (Feature-Oblivious Label DP). Consider the two-party setting where the "features party" holds as input a sequence $(x_i)_{i=1}^n$ of all feature vectors (across all the $n$ users), and the "labels party" holds as input a sequence $(y_i)_{i=1}^n$ of the corresponding labels. The labels party sends a single message $M(y_1, \ldots, y_n)$ to the features party; this message can be randomized using the internal randomness of the labels party. Based on its input and on this incoming message, the features party then trains an ML model that it outputs. We say that this output is *feature-oblivious* $\varepsilon$-LabelDP if the message $M(y_1, \ldots, y_n)$ is $\varepsilon$-DP with respect to the adjacency relation where a single $y_i$ can differ across the two datasets.

We stress the practical applicability of the setting described in Definition 3. The labels party could be an advertiser who observes the purchase data of the users, and wants to enable a publisher (the features party) to train a model predicting the likelihood of a purchase (on the advertiser) being driven by the showing of a certain type of ad (on the publisher). The advertiser would also want to limit the leakage of sensitive information to the publisher. From a practical standpoint, the simplest option for the advertiser is to send a single privatized message to the publisher who can then train a model using the features at its disposal. This is exactly the feature-oblivious label DP setting.

## 3 LABEL DP LEARNING ALGORITHM

A common template for learning with label DP is to: (i) compute noisy labels using a local DP mechanism $\mathcal{M}$, (ii) use a learning algorithm on the dataset with noisy labels. All of the baseline algorithms that we consider follow this template, through different ways of generating noisy labels, such as, (a) randomized response (for categorical labels), (b) (continuous/discrete) Laplace mechanism, (c) (continuous/discrete) staircase mechanism, etc. (for formal definitions of these mechanisms, see Appendix D). Intuitively, such a learning algorithm will be most effective when the noisy label mostly agrees with the true label.

Suppose the loss function $\ell$ satisfies the triangle inequality, namely that $\ell(\tilde{y}, y) \leq \ell(\hat{y}, y) + \ell(\tilde{y}, \hat{y})$ for all $y, \hat{y}, \tilde{y}$. Then we have that

$$\mathop{\mathbb{E}}_{(x,y)\sim\mathcal{D}} \ell(f_\theta(x), y) \leq \mathop{\mathbb{E}}_{\substack{y\sim P \\ \hat{y}\sim\mathcal{M}(y)}} \ell(\hat{y}, y) + \mathop{\mathbb{E}}_{\substack{(x,y)\sim\mathcal{D} \\ \hat{y}\sim\mathcal{M}(y)}} \ell(f_\theta(x), \hat{y}), \tag{1}$$

where, for ease of notation, we use $P$ to denote the marginal on $y$ for $(x, y) \sim \mathcal{D}$. The learning algorithm, in part (ii) of the template above, aims to minimize the second term in the RHS of (1). Thus, it is natural to choose a mechanism $\mathcal{M}$, in part (i) of the template, to minimize the first term in the RHS of (1).[3] Ghazi et al. (2021a) studied this question for the case of 0-1 loss and showed that a randomized response on top $k$ labels with the highest prior masses (for some $k$) is optimal. Their characterization was limited to this *classification* loss. In this work, we develop a characterization for a large class of *regression* loss functions.

---

[3]Note that the first term in the RHS of (1) is a constant, independent of the number of training samples. Therefore, this upper bound is not vanishing in the number of samples, and hence this inequality can be quite loose.

**Algorithm 1** RR-on-Bins$_\varepsilon^\Phi$.

**Parameters:** $\Phi : \mathcal{Y} \to \hat{\mathcal{Y}}$ (for label set $\mathcal{Y}$ and output set $\hat{\mathcal{Y}}$), privacy parameter $\varepsilon \geq 0$.
**Input:** A label $y \in \mathcal{Y}$.
**Output:** $\hat{y} \in \hat{\mathcal{Y}}$.

**return** a sample $\hat{y} \sim \hat{Y}$, where the random variable $\hat{Y}$ is distributed as

$$\Pr[\hat{Y} = \hat{y}] = \begin{cases} \frac{e^\varepsilon}{e^\varepsilon + |\hat{\mathcal{Y}}| - 1} & \text{if } \hat{y} = \Phi(y) \\ \frac{1}{e^\varepsilon + |\hat{\mathcal{Y}}| - 1} & \text{otherwise,} \end{cases}$$

for each $\hat{y} \in \hat{\mathcal{Y}}$

**Algorithm 2** Compute optimal $\Phi$ for RR-on-Bins$_\varepsilon^\Phi$.

**Input:** Distribution $P$ over $\mathcal{Y} \subseteq \mathbb{R}$, privacy param. $\varepsilon \geq 0$, loss function $\ell : \mathbb{R}^2 \to \mathbb{R}_{\geq 0}$.
**Output:** Output set $\hat{\mathcal{Y}} \subseteq \mathbb{R}$ and $\Phi : \mathcal{Y} \to \hat{\mathcal{Y}}$.

$y^1, \ldots, y^k \leftarrow$ elements of $\mathcal{Y}$ in increasing order
Initialize $A[i][j] \leftarrow \infty$ for all $i, j \in \{0, \ldots, k\}$
**for** $r, i \in \{1, \ldots, k\}$ **do**
    $L[r][i] \leftarrow \min_{\hat{y} \in \mathbb{R}} \sum_{y \in \mathcal{Y}} p_y \cdot e^{\mathbb{1}[y \in [y^r, y^i]] \cdot \varepsilon} \cdot \ell(\hat{y}, y)$
$A[0][0] \leftarrow 0$
**for** $i \in \{1, \ldots, k\}$ **do**
    **for** $j \in \{1, \ldots, i\}$ **do**
        $A[i][j] \leftarrow \min_{0 \leq r < i} A[r][j-1] + L[r+1][i]$
**return** $\Phi, \hat{\mathcal{Y}}$ correspond. to $\min_{d \in [k]} \frac{1}{d - 1 + e^\varepsilon} A[k][d]$

## 3.1 RANDOMIZED RESPONSE ON BINS: AN OPTIMAL MECHANISM

We propose a new mechanism for generating noisy labels that minimizes the first term in the RHS of (1). Namely, we define *randomized response on bins* (RR-on-Bins$_\varepsilon^\Phi$), which is a randomized algorithm parameterized by a scalar $\varepsilon > 0$ and a *non-decreasing* function $\Phi : \mathcal{Y} \to \hat{\mathcal{Y}}$ that maps the label set $\mathcal{Y} \subseteq \mathbb{R}$ to an output set $\hat{\mathcal{Y}} \subseteq \mathbb{R}$. This algorithm is simple: perform $\varepsilon$-randomized response on $\Phi(y)$, randomizing over $\hat{\mathcal{Y}}$; see Algorithm 1 for a formal definition. Any $\Phi$ we consider will be non-decreasing unless otherwise stated, and so we often omit mentioning it explicitly.

An important parameter in RR-on-Bins is the mapping $\Phi$ and the output set $\hat{\mathcal{Y}}$. We choose $\Phi$ with the goal of minimizing $\mathbb{E}\,\ell(\hat{y}, y)$. First we show, under some basic assumptions about $\ell$, that for any given distribution $P$ over labels $\mathcal{Y}$, there exists a non-decreasing map $\Phi$ such that $\mathcal{M} = \text{RR-on-Bins}_\varepsilon^\Phi$ minimizes $\mathbb{E}_{y \sim P, \hat{y} \sim \mathcal{M}(y)}\,\ell(\hat{y}, y)$. Since $\Phi$ is non-decreasing, it follows that $\Phi^{-1}(\hat{y})$ is an interval[4] of $\mathcal{Y}$, for all $\hat{y} \in \hat{\mathcal{Y}}$. Exploiting this property, we give an efficient algorithm for computing the optimal $\Phi$.

To state our results formally, we use $\mathcal{L}(\mathcal{M}; P)$ to denote $\mathbb{E}_{y \sim P, \hat{y} \sim \mathcal{M}(y)}\,\ell(\hat{y}, y)$, the first term in the RHS of (1). Our results hold under the following natural assumption; note that all the standard loss functions such as squared loss, absolute-value loss, and Poisson log loss satisfy Assumption 4.

**Assumption 4.** *Loss function $\ell : \mathbb{R} \times \mathbb{R} \to \mathbb{R}_{\geq 0}$ is such that*

- *For all $y \in \mathbb{R}$, $\ell(\cdot, y)$ is continuous.*
- *For all $y \in \mathbb{R}$, $\ell(\hat{y}, y)$ is decreasing in $\hat{y}$ when $\hat{y} \leq y$ and increasing in $\hat{y}$ when $\hat{y} \geq y$.*
- *For all $\hat{y} \in \mathbb{R}$, $\ell(\hat{y}, y)$ is decreasing in $y$ when $y \leq \hat{y}$ and increasing in $y$ when $y \geq \hat{y}$.*

We can now state our main result:

**Theorem 5.** *For any loss function $\ell : \mathbb{R} \times \mathbb{R} \to \mathbb{R}_{\geq 0}$ satisfying Assumption 4, all finitely supported distributions $P$ over $\mathcal{Y} \subseteq \mathbb{R}$, there is an output set $\hat{\mathcal{Y}} \subseteq \mathbb{R}$ and a non-decreasing map $\Phi : \mathcal{Y} \to \hat{\mathcal{Y}}$ such that*

$$\mathcal{L}(\text{RR-on-Bins}_\varepsilon^\Phi; P) \;=\; \inf_{\mathcal{M}} \mathcal{L}(\mathcal{M}; P),$$

*where the infimum is over all $\varepsilon$-DP mechanisms $\mathcal{M}$.*

*Proof Sketch.* This proof is done in two stages. First, we handle the case where $\hat{\mathcal{Y}}$ is restricted to be a subset of $\mathcal{O}$, where $\mathcal{O}$ is a finite subset of $\mathbb{R}$. Under this restriction, the optimal mechanism $\mathcal{M}$ that minimizes $\mathcal{L}(\mathcal{M}; \mathcal{P})$ can be computed as the solution to an LP. Since an optimal solution to an

---

[4]An *interval* of $\mathcal{Y}$ is a subset of the form $[a, b] \cap \mathcal{Y}$, for some $a, b \in \mathbb{R}$, consisting of consecutive elements on $\mathcal{Y}$ in sorted order.

---

**Algorithm 3** Labels Party's Randomizer LabelRandomizer$_{\varepsilon_1, \varepsilon_2}$.

---

**Parameters:** Privacy parameters $\varepsilon_1, \varepsilon_2 \geq 0$.
**Input:** Labels $y_1, \ldots, y_n \in \mathcal{Y}$.
**Output:** $\hat{y}_1, \ldots, \hat{y}_n \in \hat{\mathcal{Y}}$.

$P' \leftarrow \mathcal{M}_{\varepsilon_1}^{\mathrm{Lap}}(y_1, \ldots, y_n)$
$\Phi', \hat{\mathcal{Y}}' \leftarrow$ Result of running Algorithm 2 with distribution $P'$ and privacy parameter $\varepsilon_2$
**for** $i \in [n]$ **do**
    $\hat{y}_i \leftarrow$ RR-on-Bins$_{\varepsilon_2}^{\Phi'}(y_i)$
**return** $(\hat{y}_1, \ldots, \hat{y}_n)$

---

LP can always be found at the vertices of the constraint polytope, we study properties of the vertices of the said polytope and show that the minimizer amongst its vertices necessarily takes the form of RR-on-Bins$_{\varepsilon}^{\Phi}$. To handle the case of general $\hat{\mathcal{Y}}$, we first note that due to Assumption 4, it suffices to consider $\hat{\mathcal{Y}} \subseteq [y_{\min}, y_{\max}]$ (where $y_{\min} = \min_{y \in \mathcal{Y}} y$ and $y_{\max} = \max_{y \in \mathcal{Y}} y$). Next, we consider a sequence of increasingly finer discretizations of $[y_{\min}, y_{\max}]$, and show that RR-on-Bins$_{\varepsilon}^{\Phi}$ can come arbitrarily close to the optimal $\mathcal{L}(\mathcal{M}; P)$ when restricting $\hat{\mathcal{Y}}$ to be a subset of the discretized set. But observe that any $\Phi$ induces a partition of $\mathcal{Y}$ into intervals. Since there are only finitely many partitions of $\mathcal{Y}$ into intervals, it follows that in fact RR-on-Bins$_{\varepsilon}^{\Phi}$ can exactly achieve the optimal $\mathcal{L}(\mathcal{M}; P)$. The full proof is deferred to Appendix A.   □

Given Theorem 5, it suffices to focus on RR-on-Bins$_{\varepsilon}^{\Phi}$ mechanisms and optimize over the choice of $\Phi$. We give an efficient dynamic programming-based algorithm for the latter in Algorithm 2. The main idea is as follows. We can breakdown the representation of $\Phi$ into two parts (i) a partition $\mathcal{P}_{\Phi} = \{S_1, S_2, \ldots\}$ of $\mathcal{Y}$ into intervals[5], such that $\Phi$ is constant over each $S_i$, and (ii) the values $y_i$ that $\Phi(\cdot)$ takes over interval $S_i$. Let $y^1, \ldots, y^k$ be elements of $\mathcal{Y}$ in increasing order. Our dynamic program has a state $A[i][j]$, which is (proportional to) the optimal mechanism if we restrict the input to only $\{y^1, \ldots, y^i\}$ and consider partitioning into $j$ intervals $S_1, \ldots, S_j$. To compute $A[i][j]$, we try all possibilities for $S_j$. Recall that $S_j$ must be an interval, i.e., $S_j = \{y^{r+1}, \ldots, y^i\}$ for some $r < i$. This breaks the problem into two parts: computing optimal RR-on-Bins for $\{y^1, \ldots, y^r\}$ for $j - 1$ partitions and solving for the optimal output label $\hat{y}$ for $S_j$. The answer to the first part is simply $A[r][j-1]$, whereas the second part can be written as a univariate optimization problem (denoted by $L[r][i]$ in Algorithm 2). When $\ell$ is convex (in the first argument), this optimization problem is convex and can thus be solved efficiently. Furthermore, for squared loss, absolute-value loss, and Poisson log loss, this problem can be solved in amortized $O(1)$ time, resulting in a total running time of $O(k^2)$ for the entire dynamic programming algorithm. The full description is presented in Algorithm 2[6]; the complete analyses of its correctness and running time are deferred to Appendix B.

### 3.2 ESTIMATING THE PRIOR PRIVATELY

In the previous subsection, we assumed that the distribution $P$ of labels is known beforehand. This might not be the case in all practical scenarios. Below we present an algorithm that first privately approximates the distribution $P$ and work with this approximation instead. We then analyze how using such an approximate prior affects the performance of the algorithm.

To formalize this, let us denote by $\mathcal{M}_{\varepsilon}^{\mathrm{Lap}}$ the $\varepsilon$-DP Laplace mechanism for approximating the prior. Given $n$ samples drawn from $P$, $\mathcal{M}_{\varepsilon}^{\mathrm{Lap}}$ constructs a histogram over $\mathcal{Y}$ and adds Laplace noise with scale $2/\varepsilon$ to each entry, followed by clipping (to ensure that entries are non-negative) and normalization. A formal description of $\mathcal{M}_{\varepsilon}^{\mathrm{Lap}}$ can be found in Algorithm 4 in the Appendix.

---

[5]For convenience, we assume that $S_1, S_2, \ldots$ are sorted in increasing order.

[6]We remark that the corresponding $\Phi, \hat{\mathcal{Y}}$ on the last line can be efficiently computed by recording the minimizer for each $A[i][j]$ and $L[r][i]$, and going backward starting from $i = k$ and $j = \arg\min_{d \in [k]} \frac{1}{d-1+e^{\varepsilon}} A[k][d]$.

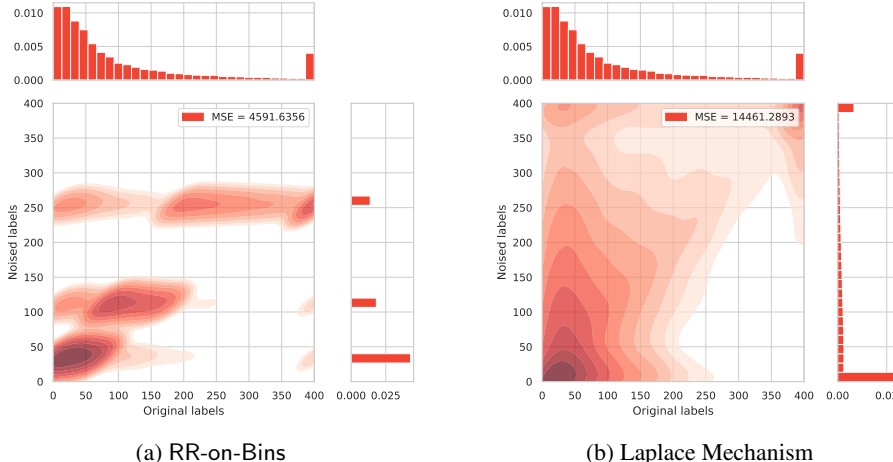

| (a) RR-on-Bins | (b) Laplace Mechanism |

Figure 2: Illustration of the training label randomization mechanism under privacy budget $\varepsilon = 3$. In this case, RR-on-Bins maps the labels to 3 bins chosen for optimal MSE via Algorithm 2. The 2D density plot contours are generated in log scale. The legends show the MSE between the original labels and the $\varepsilon$-DP randomized labels.

Our mechanism for the unknown prior case—described in Algorithm 3—is now simple: We split the privacy budget into $\varepsilon_1, \varepsilon_2$, run the $\varepsilon_1$-DP Laplace mechanism to get an approximate prior distribution $P'$ and run the optimal RR-on-Bins for privacy parameter $\varepsilon_2$ to randomize the labels. It is not hard to see that the algorithm is $(\varepsilon_1 + \varepsilon_2)$-DP. (Full proof deferred to the Appendix.)

**Theorem 6.** LabelRandomizer$_{\varepsilon_1,\varepsilon_2}$ *is* $(\varepsilon_1 + \varepsilon_2)$-*DP.*

Let us now discuss the performance of the above algorithm in comparison to the case where the prior $P$ is known, under the assumption that $y_1, \ldots, y_n$ are sampled i.i.d. from $P$. We show in the following theorem that the difference between the expected population loss in the two cases converges to zero as the number of samples $n \to \infty$, provided that $\mathcal{Y}$ is finite:

**Theorem 7.** *Let* $\ell : \mathbb{R} \times \mathbb{R} \to \mathbb{R}_{\geq 0}$ *be any loss function satisfying Assumption 4. Furthermore, assume that* $\ell(\hat{y}, y) \leq B$ *for some parameter $B$ for all* $y, \hat{y} \in \mathcal{Y}$. *For any distribution $P$ on $\mathcal{Y}$, $\varepsilon > 0$, and any sufficiently large $n \in \mathbb{N}$, there is a choice of $\varepsilon_1, \varepsilon_2 > 0$ such that $\varepsilon_1 + \varepsilon_2 = \varepsilon$ and*

$$\mathop{\mathbb{E}}_{\substack{y_1,\ldots,y_n \sim P \\ P', \Phi', \hat{\mathcal{Y}}'}} [\mathcal{L}(\text{RR-on-Bins}_{\varepsilon_2}^{\Phi'}; P)] - \inf_{\mathcal{M}} \mathcal{L}(\mathcal{M}; P) \leq O\left(B \cdot \sqrt{|\mathcal{Y}|/n}\right),$$

*where* $P', \Phi', \hat{\mathcal{Y}}'$ *are as in* LabelRandomizer$_{\varepsilon_1,\varepsilon_2}$ *and the infimum is over all $\varepsilon$-DP mechanisms $\mathcal{M}$.*

We remark that the above bound is achieved by setting $\varepsilon_1 = \sqrt{|\mathcal{Y}|/n}$; indeed, we need to assume that $n$ is sufficiently large so that $\varepsilon_1 < \varepsilon$. The high-level idea of the proof is to first use a known utility bound for Laplace mechanism Diakonikolas et al. (2015) to show that $P, P'$ are close. Then, we argue that the optimal RR-on-Bins is robust, in the sense that small changes in the prior (from $P$ to $P'$) and privacy parameter (from $\varepsilon$ to $\varepsilon_2$) do not affect the resulting population loss too much.

## 4 EXPERIMENTAL EVALUATION

We evaluate the RR-on-Bins mechanism on three datasets, and compare with the Laplace mechanism (Dwork et al., 2006b), the staircase mechanism (Geng & Viswanath, 2014) and the exponential mechanism (McSherry & Talwar, 2007). Note the Laplace mechanism and the staircase mechanism both have a discrete and a continuous variant. For real-valued labels (the Criteo Sponsored Search Conversion dataset), we use the continuous variant, and for integer-valued labels (the US Census dataset and the App Ads Conversion Count dataset), we use the discrete variant. All of these algorithms can be implemented in the feature-oblivious label DP setting of Figure 1. Detailed model and training configurations can be found in Appendix E.

| Privacy Budget | MSE (Mechanism) | | MSE (Generalization) | |
|---|---|---|---|---|
| | Laplace Mechanism | RR-on-Bins | Laplace Mechanism | RR-on-Bins |
| 0.05 | $60\,746.98 \pm 46.31$ | $11\,334.84 \pm 9.07$ | $24\,812.56 \pm 139.35$ | $11\,339.71 \pm 36.45$ |
| 0.1 | $59\,038.06 \pm 51.31$ | $11\,325.53 \pm 9.25$ | $23\,933.23 \pm 172.43$ | $11\,328.04 \pm 36.34$ |
| 0.3 | $52\,756.01 \pm 56.64$ | $11\,210.48 \pm 9.06$ | $20\,961.83 \pm 149.47$ | $11\,185.20 \pm 36.10$ |
| 0.5 | $47\,253.12 \pm 57.12$ | $10\,977.09 \pm 8.85$ | $18\,411.30 \pm 111.82$ | $10\,901.33 \pm 36.54$ |
| 0.8 | $40\,223.13 \pm 48.66$ | $10\,435.43 \pm 9.77$ | $15\,428.75 \pm 91.32$ | $10\,256.37 \pm 37.39$ |
| 1.0 | $36\,226.54 \pm 45.05$ | $9\,976.86 \pm 8.21$ | $13\,788.51 \pm 75.71$ | $9\,744.08 \pm 37.59$ |
| 1.5 | $28\,170.93 \pm 39.45$ | $8\,636.43 \pm 7.04$ | $10\,808.53 \pm 52.31$ | $8\,406.88 \pm 36.57$ |
| 2.0 | $22\,219.20 \pm 28.04$ | $7\,260.05 \pm 10.55$ | $8\,892.80 \pm 32.92$ | $7\,294.93 \pm 34.03$ |
| 3.0 | $14\,411.77 \pm 20.26$ | $4\,600.24 \pm 11.15$ | $6\,770.33 \pm 22.86$ | $5\,577.50 \pm 31.75$ |
| 4.0 | $9\,851.53 \pm 17.27$ | $2\,631.36 \pm 4.41$ | $5\,764.32 \pm 28.95$ | $4\,769.61 \pm 25.01$ |
| 6.0 | $5\,270.57 \pm 10.30$ | $709.74 \pm 6.18$ | $4\,955.21 \pm 26.75$ | $4\,371.68 \pm 25.31$ |
| 8.0 | $3\,239.22 \pm 6.54$ | $176.47 \pm 2.12$ | $4\,668.40 \pm 20.34$ | $4\,333.12 \pm 31.94$ |
| $\infty$ | $0.00 \pm 0.00$ | $0.00 \pm 0.00$ | $4\,322.91 \pm 28.31$ | $4\,319.86 \pm 29.27$ |

Table 1: MSE on the Criteo dataset. The first column block (Mechanism) measures the error introduced by the DP randomization mechanisms on the training labels. The second column block (Generalization) measures the test error of the models trained on the corresponding private labels.

## 4.1 CRITEO SPONSORED SEARCH CONVERSION

The Criteo Sponsored Search Conversion Log Dataset (Tallis & Yadav, 2018) contains logs obtained from Criteo Predictive Search (CPS). Each data point describes an action performed by a user (click on a product related advertisement), with additional information consisting of a conversion (product was bought) within a 30-day window and that could be attributed to the action. We formulate a (feature-oblivious) label DP problem to predict the revenue (in €) obtained when a conversion takes place (the SalesAmountInEuro attribute). This dataset represents a sample of 90 days of Criteo live traffic data, with a total of $15,995,634$ examples. We remove examples where no conversion happened (SalesAmountInEuro is $-1$), resulting in a dataset of $1,732,721$ examples. The conversion value goes up to $62,458.773$€. We clip the conversion value at $400$€, which corresponds to the 95th percentile of the value distribution.

In Figure 2, we visualize an example of how RR-on-Bins randomizes those labels, and compare it with the Laplace mechanism, which is a canonical DP mechanism for scalars. In this case (and for $\varepsilon = 3$), RR-on-Bins chooses 3 bins at around $50$, $100$, and $250$ and maps the sensitive labels to those bins with Algorithm 1. The joint distribution of the sensitive labels and randomized labels maintains an overall concentration along the diagonal. On the other hand, the joint distribution for the Laplace mechanism is generally spread out. Table 1 quantitatively compares the two mechanisms across different privacy budgets. The first block (Mechanism) shows the MSE between the sensitive *training* labels and the private labels generated by the two mechanisms, respectively. We observe that RR-on-Bins leads to significantly smaller MSE than the Laplace mechanism for the same label DP guarantee. Furthermore, as shown in the second block of Table 1 (Generalization), the reduced noise in the training labels leads to lower *test* errors.

Figure 3 compares with two additional baselines: the exponential mechanism, and the staircase mechanism. For both the "Mechanism" errors and "Generalization" errors, RR-on-Bins consistently outperforms other methods for both low- and high-$\varepsilon$ regimes.

## 4.2 US CENSUS

The 1940 US Census dataset has been made publicly available for research since 2012, and has $131,903,909$ rows. This data is commonly used in the DP literature (e.g., Wang et al. (2019); Cao et al. (2021); Ghazi et al. (2021b)). In this paper, we set up a label DP problem by learning to predict the duration for which the respondent worked during the previous year (the WKSWORK1 field, measured in number of weeks). Figure 4a shows that RR-on-Bins outperforms the baseline mechanisms across a wide range of privacy budgets.

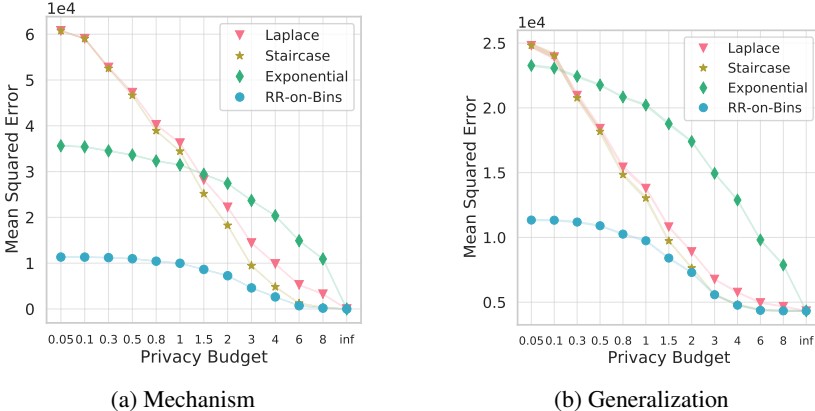

(a) Mechanism           (b) Generalization

Figure 3: MSE on the Criteo dataset: (a) error introduced by DP randomization mechanisms on the training labels and (b) test error of the models trained on the corresponding private labels.

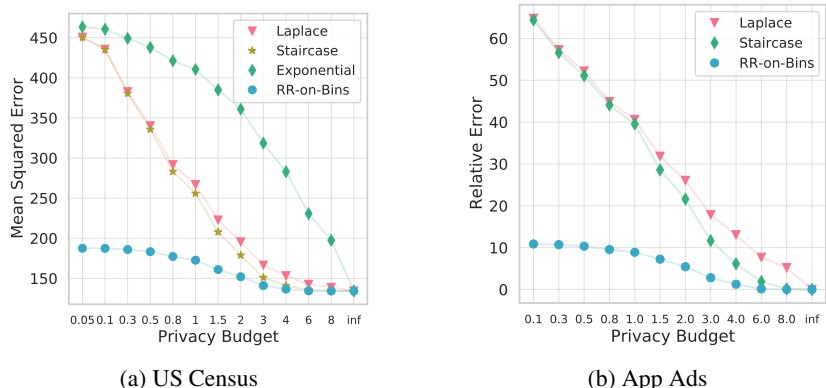

(a) US Census           (b) App Ads

Figure 4: Test performance across different privacy budgets. (a) MSE on the US Census dataset. (b) Relative performance on the App Ads Conversion Count dataset. The relative error is calculated with respect to the non-private baseline, i.e. $(E - E_{\text{baseline}})/E_{\text{baseline}}$.

### 4.3 APP ADS CONVERSION COUNT PREDICTION

Finally, we evaluate our algorithm on an app install ads prediction dataset from a commercial mobile app store. The examples in this dataset are ad clicks and each label counts post-click events (aka conversions) occurring in the app after the user installs it. For example, if a user installs a ride share app after clicking the corresponding ad, the label could be the total number of rides that the user purchases in a given time window after the installation. We note that ad prediction tasks/datasets of a similar nature have been previously used for evaluation (Badanidiyuru et al., 2021).

Figure 4b shows the performance on this dataset. Consistent with the results observed on the other datasets, RR-on-Bins outperforms other mechanisms across all the privacy budget values.

In summary, by first estimating a prior distribution (privately), RR-on-Bins significantly reduces label noise compared to other label DP mechanisms, especially in the high-privacy (i.e., intermediate to small $\varepsilon$) regime. This leads to significantly better test performance for the regression networks trained on these less noisy labels. For example, for $\varepsilon = 0.5$, comparing to the best baseline methods, the test MSE for RR-on-Bins is $\sim 1.5\times$ smaller on the Criteo and US Census datasets, and the relative test error is $\sim 5\times$ smaller on the App Ads dataset.

## 5   RELATED WORK

There has been a large body of work on learning with DP. Various theoretical and practical settings have been studied including empirical risk minimization (Chaudhuri et al., 2011), optimization (Song et al., 2013), regression analysis (Zhang et al., 2012), and deep learning (Abadi et al., 2016). The vast majority of the prior works studied the setting where both the features and the labels are deemed sensitive and hence ought to be protected.

Chaudhuri & Hsu (2011) and Beimel et al. (2016) studied the sample complexity of learning with label DP for classification tasks. Wang & Xu (2019) studied label DP for sparse linear regression in the local DP setting. Ghazi et al. (2021a) provided a procedure for training deep neural classifiers with label DP. For the classification loss, they formulate an LP and derive an explicit solution, which they refer to as RRTop-$k$. Our work could be viewed as an analog of theirs for regression tasks. Malek Esmaeili et al. (2021) used the PATE framework of Papernot et al. (2017; 2018) to propose a new label DP training method. We note that this, as well as related work on using unsupervised and semi-supervised learning to improve label DP algorithms (Esfandiari et al., 2022; Tang et al., 2022), do not apply to the feature-oblivious label DP setting.

A variant of two-party learning (with a "features party" and a "labels party") was recently studied by Li et al. (2021), who considered the interactive setting where the two parties can engage in a protocol with an arbitrary number of rounds (with the same application to computational advertising described in Section 1 as a motivation). By contrast, we considered in this paper the one-way communication setting (from Figure 1), which is arguably more practical and easier to deploy.

Kairouz et al. (2016) study optimal local DP algorithms under a given utility function and prior. For a number of utility functions, they show that optimal mechanisms belong to a class of *staircase mechanisms*. Staircase mechanisms are much more general than randomized response on bins. In that regard, our work shows that a particular subset of staircase mechanisms is optimal for regression tasks. In particular, while we give an efficient dynamic programming algorithm for optimizing over randomized response on bins, we are not aware of any efficient algorithm that can compute an optimal staircase mechanism. (In particular, a straightforward algorithm takes $2^{O(k)}$ time.)

## 6   CONCLUSIONS AND FUTURE DIRECTIONS

In this work we propose a new label DP mechanism for regression. The resulting training algorithm can be implemented in the feature-oblivious label DP setting. We provide theoretical results shedding light on the operation and guarantees of the algorithm. We also evaluate it on three datasets, demonstrating that it achieves higher accuracy compared to several non-trivial, natural approaches.

Our work raises many questions for future exploration. First, in our evaluation, we set the objective function optimized using the LP to be the same as the loss function used during training; different ways of setting the LP objective in relation to the training loss are worth exploring. Second, our noising mechanism typically introduces a bias; mitigating it, say by adding unbiasedness constraints into the LP, is an interesting direction. Third, it might also be useful to investigate de-noising techniques that can be applied as a post-processing step on top of our label DP mechanism; e.g., the method proposed in Tang et al. (2022) for computer vision tasks, and the ALIBI method (Malek Esmaeili et al., 2021) for classification tasks, which is based on Bayesian inference.

We believe that due to its simplicity and practical appeal, the setting of learning with feature-oblivious label DP merits further study, from both a theoretical and an empirical standpoint.

Finally, we leave the question of obtaining better "feature-aware" label DP regression algorithms for a future investigation.

### ACKNOWLEDGMENTS

We thank Peter Kairouz and Sewoong Oh for many useful discussions, and Eu-Jin Goh, Sam Ieong, Christina Ilvento, Andrew Tomkins, and the anonymous reviewers for their comments.

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

## A  OPTIMALITY OF RR-on-Bins$_\varepsilon^\Phi$

In this section, we prove Theorem 5. We prove this in two stages. First we consider the case where $P$ has finite support and $\hat{\mathcal{Y}}$ is constrained to be a subset of a pre-specified finite set $\mathcal{O} \subseteq \mathbb{R}$. Next, we consider the case where $\hat{\mathcal{Y}}$ is allowed to be an arbitrary subset of $\mathbb{R}$, and thus, the output distribution of the mechanism on any input $y$ can be an arbitrary probability measure over $\mathbb{R}$.

**Stage I: Finitely supported $P$, $\hat{\mathcal{Y}} \subseteq \mathcal{O}$, $\ell$ strictly increasing/decreasing.**

**Theorem 8.** *Let $P$ be a distribution over $\mathbb{R}$ with finite support, $\mathcal{O}$ a finite subset of $\mathbb{R}$, and $\ell : \mathbb{R} \times \mathbb{R} \to \mathbb{R}_{\geq 0}$ be such that*

- *For all $y \in \mathbb{R}$, $\ell(\hat{y}, y)$ is strictly decreasing in $\hat{y}$ when $\hat{y} \leq y$ and strictly increasing in $\hat{y}$ when $\hat{y} \geq y$.*
- *For all $\hat{y} \in \mathbb{R}$, $\ell(\hat{y}, y)$ is strictly decreasing in $y$ when $y \leq \hat{y}$ and strictly increasing in $y$ when $y \geq \hat{y}$.*

*Then for all $\varepsilon > 0$, there exists $\Phi : \mathcal{Y} \to \mathcal{O}$ such that*

$$\mathcal{L}(\text{RR-on-Bins}_\varepsilon^\Phi; P) \;=\; \inf_{\mathcal{M}} \mathcal{L}(\mathcal{M}; P).$$

*where $\inf_{\mathcal{M}}$ is over all $\varepsilon$-DP mechanisms $\mathcal{M}$ with inputs in $\mathcal{Y}$ and outputs in $\mathcal{O}$.*

The proof of Theorem 8 is inspired by the proof of Theorem 3.1 of Ghosh et al. (2012). Namely, we describe the problem of minimizing $\mathcal{L}(\mathcal{M}; P)$ as a linear program (LP). We arrange the variables of the LP into a matrix and associate any solution to the LP a signature matrix, which represents when constraints of the LP are met with equality. Then we make several observations of the signature matrix for any optimal solution, which eventually leads to the proof of the theorem.

*Proof of Theorem 8.* Without loss of generality, we may assume that $\mathcal{Y} = \text{supp}(P)$ and therefore is finite.

When $\hat{\mathcal{Y}}$ is restricted to be a subset of $\mathcal{O}$, the optimal mechanism $\mathcal{M}$ which minimizes $\mathcal{L}(\mathcal{M}; P) = \mathbb{E}_{y \sim P, \hat{y} \sim \mathcal{M}(y)}\, \ell(\hat{y}, y)$, is encoded in the solution of the following LP, with $|\mathcal{Y}| \cdot |\mathcal{O}|$ variables $M_{y \to \hat{y}} = \Pr[\mathcal{M}(y) = \hat{y}]$ indexed by $(y, \hat{y}) \in \mathcal{Y} \times \mathcal{O}$. Namely, the first two constraints enforce that $(M_{y \to \hat{y}})_y$ is a valid probability distribution and the third constraint enforces the $\varepsilon$-DP constraint.

$$
\begin{aligned}
\min_{M} \quad & \sum_{y \in \mathcal{Y}} p_y \left( \sum_{\hat{y} \in \mathcal{O}} M_{y \to \hat{y}} \cdot \ell(\hat{y}, y) \right), \\
\text{subject to} \quad & \forall y \in \mathcal{Y},\, \hat{y} \in \mathcal{O} : & M_{y \to \hat{y}} \geq 0, \\
& \forall y \in \mathcal{Y} : & \sum_{\hat{y} \in \mathcal{O}} M_{y \to \hat{y}} = 1, \\
& \forall \hat{y} \in \mathcal{O}, \forall y, y' \in \mathcal{Y}, y \neq y' : & M_{y' \to \hat{y}} \leq e^\varepsilon \cdot M_{y \to \hat{y}}.
\end{aligned}
\tag{2}
$$

Corresponding to any feasible solution $M := (M_{y \to \hat{y}})_{y, \hat{y}}$, we associate a $|\mathcal{Y}| \times |\mathcal{O}|$ *signature* matrix $S_M$. First, let $p_{\hat{y}}^{\min} = \min_y M_{y \to \hat{y}}$ and let $p_{\hat{y}}^{\max} = \max_y M_{y \to \hat{y}}$. Note that, from the constraints it follows that $p_{\hat{y}}^{\max} \leq e^\varepsilon \cdot p_{\hat{y}}^{\min}$.

**Definition 9** (Signature matrix). For any feasible $M$ for the LP in (2), the *signature* entry $S_M(y, \hat{y})$ for all $y \in \mathcal{Y}$ and $\hat{y} \in \mathcal{O}$ is defined as

$$
S_M(y, \hat{y}) = \begin{cases}
0 & \text{if } M_{y \to \hat{y}} = 0 \\
\mathsf{U} & \text{if } M_{y \to \hat{y}} = p_{\hat{y}}^{\max} = e^\varepsilon \cdot p_{\hat{y}}^{\min} \\
\mathsf{L} & \text{if } M_{y \to \hat{y}} = p_{\hat{y}}^{\min} = e^{-\varepsilon} \cdot p_{\hat{y}}^{\max} \\
\mathsf{S} & \text{otherwise.}
\end{cases}
$$

| | 1.5 | 2.5 | 3.5 | 4.5 |
|---|---|---|---|---|
| 1 | 0 | $1/2$ | $1/4$ | $1/4$ |
| 2 | 0 | $1/2$ | $1/4$ | $1/4$ |
| 3 | 0 | $1/4$ | $1/4$ | $1/2$ |
| 4 | 0 | $1/3$ | $1/3$ | $1/3$ |
| 5 | 0 | $1/3$ | $1/3$ | $1/3$ |

$$M$$

| | 1.5 | 2.5 | 3.5 | 4.5 |
|---|---|---|---|---|
| 1 | 0 | U | S | L |
| 2 | 0 | U | S | L |
| 3 | 0 | L | S | U |
| 4 | 0 | S | S | S |
| 5 | 0 | S | S | S |

$$S_M$$

Figure 5: Example of a signature matrix for $e^\varepsilon = 1/2$ for $\mathcal{Y} = \{1, 2, 3, 4, 5\}$ and $\mathcal{O} = \{1.5, 2.5, 3.5, 4.5\}$

We visualize $S_M$ as a matrix with rows corresponding to $y$'s and columns corresponding to $\hat{y}$'s, both ordered in increasing order (see Figure 5).

If $\mathcal{M}$ is an RR-on-Bins$_\varepsilon^\Phi$ mechanism for some $\Phi : \mathcal{Y} \to \mathcal{O}$, then it is easy to see that the corresponding signature matrix satisfies some simple properties (given in Claim 10 below). Interestingly, we establish a converse, thereby characterizing the signature of matrices $M$ that correspond to RR-on-Bins$_\varepsilon^\Phi$ mechanism for some $\Phi : \mathcal{Y} \to \mathcal{O}$.

**Claim 10.** $M$ *corresponds to an* RR-on-Bins$_\varepsilon^\Phi$ *mechanism for some non-decreasing* $\Phi : \mathcal{Y} \to \hat{\mathcal{Y}}$ *(for $\hat{\mathcal{Y}} \subseteq \mathcal{O}$) if and only if either*

*(1) One column consists entirely of* S*, while all other columns are entirely 0; let $\Psi(y) = \hat{y}$, where $\hat{y}$ corresponds to the unique* S *column,*

*or all of the following hold:*

*(2a) Each column in $S_M$ is either entirely 0, or entirely consisting of* U*'s and* L*'s, with at least one* U *and one* L*.*
*(2b) Each row contains a single* U *entry, with all other entries being either* L *or 0; for each $y$, we denote this unique column index of* U *by $\Psi(y)$.*
*(2c) For all $y < y'$ it holds that $\Psi(y) \le \Psi(y')$.*

*In each case, it holds that $\Phi = \Psi$.*

*Proof of Claim 10.* Suppose $M$ corresponds to RR-on-Bins$_\varepsilon^\Phi$ for some non-decreasing $\Phi$. If $\Phi$ is constant, then condition (1) holds, since all columns corresponding to $\hat{y} \notin \text{range}(\Phi)$ in $S_M$ are all 0, whereas, the only remaining column consists of all S. If $\Phi$ is not constant, then $M_{y \to \hat{y}} = e^{\varepsilon \mathbb{1}[\Phi(y) = \hat{y}]}/(e^\varepsilon + |\hat{\mathcal{Y}}| - 1)$, and hence its signature $S_M$ is such that $S_M(y, \hat{y})$ is U if $\Phi(y) = \hat{y}$, and L if $\hat{y} \in \text{range}(\Phi) \setminus \{\Phi(y)\}$, and 0 otherwise. It is easy to verify that all three conditions hold: (2a) a column corresponding to $\hat{y}$ is entirely 0 if and only if $\hat{y} \notin \text{range}(\Phi)$ and entirely consisting of U's and L's otherwise with at least one U corresponding to $y$ such that $\Phi(y) = \hat{y}$ and at least one L corresponding to $y$ such that $\Phi(y) \ne \hat{y}$ (since $\Phi$ is non-constant), (2b) Each row corresponding to $y$ has a unique U corresponding to $\hat{y} = \Psi(y) = \Phi(y)$, and (2c) For $y < y'$, it holds that $\Psi(y) \le \Psi(y')$ since $\Phi$ is non-decreasing.

To establish the converse, suppose we have that $S_M$ satisfies condition (1). Then we immediately get that $M$ corresponds to RR-on-Bins for the constant map $\Phi = \Psi$. Next, suppose $S_M$ satisfies all three conditions (2a)–(2c). Immediately, we have that $M$ is given as

$$M_{y \to \hat{y}} = \begin{cases} e^\varepsilon \cdot p_{\hat{y}}^{\min} & \text{if } S_M(y, \hat{y}) = \text{U} \\ p_{\hat{y}}^{\min} & \text{if } S_M(y, \hat{y}) = \text{L} \\ 0 & \text{if } S_M(y, \hat{y}) = 0. \end{cases}$$

Let $\hat{\mathcal{Y}}$ correspond to the set of non-zero columns of $S_M$. Since each row of $M$ corresponds to a probability distribution, we have for each $\hat{y} \in \hat{\mathcal{Y}}$ that $\sum_{\hat{y}' \in \hat{\mathcal{Y}}} p_{\hat{y}'}^{\min} + (e^\varepsilon - 1) p_{\hat{y}}^{\min} = 1$ by considering

the row corresponding to any $y \in \Psi^{-1}(\hat{y})$. Thus, we get that $p_{\hat{y}}^{\min}$ is the same for all values in $\hat{\mathcal{Y}}$, and in particular, $p_{\hat{y}}^{\min} = \frac{1}{e^\varepsilon + |\hat{\mathcal{Y}}| - 1}$, and thus, we get that the corresponding $M$ is uniquely given by

$$
M_{y \to \hat{y}} = \begin{cases} \frac{e^\varepsilon}{e^\varepsilon + |\hat{\mathcal{Y}}| - 1} & \text{if } S_M(y, \hat{y}) = \mathsf{U} \\ \frac{1}{e^\varepsilon + |\hat{\mathcal{Y}}| - 1} & \text{if } S_M(y, \hat{y}) = \mathsf{L} \\ 0 & \text{if } S_M(y, \hat{y}) = 0, \end{cases}
$$

which clearly corresponds to $\mathsf{RR\text{-}on\text{-}Bins}_\varepsilon^\Phi$ for $\Phi(y) = \Psi(y)$. We have that $\Psi$, and hence $\Phi$, is non-decreasing from condition (2c). This completes the proof of Claim 10. $\qquad\square$

Thus, to show optimality of $\mathsf{RR\text{-}on\text{-}Bins}_\varepsilon^\Phi$, it suffices to show that there exists a minimizer $M$ of the LP (2), such that $S_M$ satisfies the conditions in Claim 10.

It is well known that an optimal solution to any LP can be found at the vertices of the constraint polytope and that for a LP with $n$ variables, vertices must meet $n$ linearly independent constraints. We use this to find a submatrix of $S_M$, which will determine $M$ in its entirety.

If $M$ is a feasible point, then each column of $S_M$ is either entirely 0, or entirely consisting of $\mathsf{S}$, $\mathsf{U}$, and $\mathsf{L}$. This is necessary since if $M_{y \to \hat{y}} = 0$, then $0 \leq M_{y' \to \hat{y}} \leq e^\varepsilon M_{y \to \hat{y}} = 0$. Let $k$ denote the number of non-zero columns of $S_M$.

**Lemma 11.** *Suppose $M$ is a vertex of the LP (2) with $k$ non-zero columns of $S_M$. If $k \geq 2$, then $M$ has a $k \times k$ submatrix $M^{(k)}$ with all distinct rows, consisting of only $\mathsf{U}$'s and $\mathsf{L}$'s.*

*Proof.* $M$ is a vertex if and only if there are $|\mathcal{Y}| \cdot |\mathcal{O}|$ many linearly independent constraints that are tight. We count the number of tight constraints just from $S_M$ and note some instances of dependence to give a lower bound on the number of linearly independent constraints.

1. $|\mathcal{Y}|$ constraints, given by $\sum_{\hat{y}} M_{y \to \hat{y}} = 1$ are always tight.

2. For a zero column of $S_M$ corresponding to $\hat{y}$, each $M_{y \to \hat{y}} \geq 0$ is tight corresponding to each $y \in \mathcal{Y}$. These correspond to $|\mathcal{Y}| \cdot (|\mathcal{O}| - k)$ constraints.

3. For a non-zero column of $S_M$ corresponding to $\hat{y}$, let $C_{\hat{y}}(\mathsf{U})$ denote the number of $\mathsf{U}$ entries in column $\hat{y}$ and similarly, define $C_{\hat{y}}(\mathsf{L})$ and $C_{\hat{y}}(\mathsf{S})$ analogously. For each pair $y_1, y_2$ such that $S_M(y_1, \hat{y}) = \mathsf{L}$ and $S_M(y_2, \hat{y}) = \mathsf{U}$, we have a tight constraint $M_{y_1 \to \hat{y}} \leq e^\varepsilon \cdot M_{y_2 \to \hat{y}}$. However, these constraints are not linearly independent. In fact, there are only $C_{\hat{y}}(\mathsf{U}) + C_{\hat{y}}(\mathsf{L}) - 1 = |\mathcal{Y}| - C_{\hat{y}}(\mathsf{S}) - 1$ many linearly independent constraints among these.

4. Another instance where the constraints might be dependent is if two rows of $S_M$, corresponding to say $y_1$ and $y_2$, are identical and do not contain any $\mathsf{S}$'s. In this instance, the two equations of $\sum_{\hat{y}} M_{y_i \to \hat{y}} = 1$ and the inequalities given by the 0s, $\mathsf{U}$'s, and $\mathsf{L}$'s are not independent. The DP inequality conditions imply that all the coordinates are equal between the two rows, which imply a dependence relation between those and the two conditions.

Counting these all up, we have a lower bound on the number of linearly independent constraints. This must be at least $|\mathcal{Y}| \cdot |\mathcal{O}|$. Let (# of duplicate rows not containing $\mathsf{S}$) be the difference between the number of all rows not containing $\mathsf{S}$ and the number of distinct rows not containing $\mathsf{S}$. Thus, we get

$$
|\mathcal{Y}| + |\mathcal{Y}| \cdot (|\mathcal{O}| - k) + (|\mathcal{Y}| - 1) \cdot k - \sum_{\hat{y}} C_{\hat{y}}(\mathsf{S}) - (\# \text{ of duplicate rows not containing } \mathsf{S}) \geq |\mathcal{Y}| \cdot |\mathcal{O}|.
$$

Rearranging,

$$
|\mathcal{Y}| - \sum_{\hat{y}} C_{\hat{y}}(\mathsf{S}) - (\# \text{ of duplicate rows not containing } \mathsf{S}) \geq k,
$$

and because

$$
\sum_{\hat{y}} C_{\hat{y}}(\mathsf{S}) \geq (\# \text{ rows containing } \mathsf{S}),
$$

we conclude

$$|\mathcal{Y}| - (\#\text{ rows containing }\mathsf{S}) - (\#\text{ of duplicate rows not containing }\mathsf{S}) \geq k.$$

Hence, there are at least $k$ rows, which are all distinct and contain only 0's, U's, and L's. Narrowing our scope to just the $k$ non-zero columns, we get a $k \times k$ sub-matrix $M^{(k)}$ that contains only U's and L's. This concludes the proof of Lemma 11. $\square$

So far, we did not use any information about the objective. Next we use the properties of the loss function $\ell$ to show that if $M$ is an optimal solution to the LP, any submatrix provided by Lemma 11 can only be of one form, the matrix with U's along the diagonal and L's everywhere else:

**Lemma 12.** *Suppose $M$ is an optimal vertex solution to the LP 2 with $k \geq 2$ non-zero columns. Then any $k \times k$ submatrix $M^{(k)}$ given by Lemma 11 has U's along the diagonal and L's otherwise.*

*Proof.* Firstly, in each row of $M^{(k)}$, the U's are consecutive. Suppose for any $y$ and $\hat{y}_1 < \hat{y}_2 < \hat{y}_3$, it holds that $S_M(y, \hat{y}_1) = S_M(y, \hat{y}_3) = \mathsf{U}$, whereas $S_M(y, \hat{y}_2)$ is either L or S. This implies that $\ell(\hat{y}_1, y), \ell(\hat{y}_3, y) < \ell(\hat{y}_2, y)$, since otherwise, it is possible to reduce $M_{y \to \hat{y}_1}$ (resp. $M(y \to \hat{y}_3)$) and increase $M_{y \to \hat{y}_2}$ thereby further reducing the objective, without violating any constraints. But this is a contradiction to the assumption of $\ell$ in Theorem 8, since $\ell(\cdot, y)$ cannot increase from $\hat{y}_1$ to $\hat{y}_2$ and then decrease from $\hat{y}_2$ to $\hat{y}_3$.

Secondly, $S_M$ cannot contain the following $2 \times 2$ matrix, where $y_1 < y_2$ and $\hat{y}_1 < \hat{y}_2$:

| | $\hat{y}_1$ | $\hat{y}_2$ |
|---|---|---|
| $y_1$ | L | U |
| $y_2$ | U | L |

Since $M$ is optimal, we get that $\ell(\hat{y}_1, y_1) > \ell(\hat{y}_2, y_1)$, and from the assumption on $\ell$ it follows that $\hat{y}_1 < y_1$ (since otherwise, $y_1 \leq \hat{y}_1 < \hat{y}_2$ which would imply $\ell(\hat{y}_1, y_1) < \ell(\hat{y}_2, y_1)$). Similarly, we have that $\ell(\hat{y}_2, y_2) > \ell(\hat{y}_1, y_2)$ and $y_2 < \hat{y}_2$. However, since $\hat{y}_1 < y_1 < y_2 < \hat{y}_2$, we have that $\ell(\hat{y}_2, y_1) > \ell(\hat{y}_2, y_2)$ and $\ell(\hat{y}_1, y_2) > \ell(\hat{y}_1, y_1)$, which gives rise to a contradiction:

$$\begin{aligned}
\ell(\hat{y}_1, y_1) &> \ell(\hat{y}_2, y_1) \\
&> \ell(\hat{y}_2, y_2) \\
&> \ell(\hat{y}_1, y_2) \\
&> \ell(\hat{y}_1, y_1).
\end{aligned}$$

In particular, this claim of not containing the above $2 \times 2$ matrix applies to $M^{(k)}$.

Lastly, every row of $M^{(k)}$ has at least one $U$. Suppose for contradiction that the row $y$ in $S_M$ does not contain a single U. Then it has all L's and 0's. Any column that contains an L must contain a U in $S_M$. So necessarily there is a row $y'$ in $S_M$ containing a U. Now, $y$ containing only L's and 0's implies that $\sum_{\hat{y}} M_{y \to \hat{y}} < \sum_{\hat{y}} M_{y' \to \hat{y}}$, which is a contradiction of the constraints of the LP.

Since the rows of this $k \times k$ sub-matrix $M^{(k)}$ are all distinct, and it does not contain any $2 \times 2$ submatrix as above, the only possible signature is where all the diagonal signature entries are U and the rest are L. Let $s_i \in [k]$ be the index of the first index U for the $i$th row and similarly $e_i \in [k]$ be the last index of U. Because the U's are continuous, $s_i$ and $e_i$ determine the signature of the row. If $s_i = s_j$ for $i \neq j$, then $e_i = e_j$. Else if $e_i < e_j$, then $\sum_{\hat{y}} M_{y \to \hat{y}} < \sum_{\hat{y}} M_{y' \to \hat{y}}$ where $y$ corresponds to row $i$ and $y'$ corresponds to $j$ which is a contradiction. Similarly for $e_i > e_j$. The same argument can be made if $e_i = e_j$, then $s_i = s_j$. The rows of $M^{(k)}$ are distinct, so this implies that $s_i \neq s_j, e_i \neq e_j$ for $i \neq j$. The observation about the $2 \times 2$ matrix shows that if $i < j$, $s_i \leq s_j$ and $e_i \leq e_j$. So the $s_i$ and $e_i$ are both $k$ distinct ordered sequences of $[k]$. The only such sequence is $s_i = e_i = i$, implying $M^{(k)}$ is a diagonal matrix of this form. This completes the proof of Lemma 12. $\square$

We now have the necessary observations to prove Theorem 8. Let $M$ be an optimal vertex solution to the LP 2. That is $\mathcal{L}(M; P) = \inf_{\mathcal{M}} \mathcal{L}(\mathcal{M}; P)$. If $M$ has one non-zero column, then $S_M$ has one column entirely of S and the rest 0. By Claim 10, $M$ corresponds to a RR-on-Bins$_{\varepsilon}^{\Phi}$ mechanism. If

$M$ has $k \geq 2$ non-zero columns, then by Lemma 12, $S_M$ has a $k \times k$ submatrix $M^{(k)}$ with U along the diagonal and L otherwise. We show this completely determines $S_M$ in the form described by Claim 10(2a–2c).

Note that $S_M$ for any vertex $M$ has at most one S per row. If $S_M(y, \hat{y}_1)$ and $S_M(y, \hat{y}_2)$ are both equal to S for $y_1 \neq y_2$, then it is possible to write $M$ as a convex combination of two other feasible points given as $M + \eta M'$ and $M - \eta M'$ for small enough $\eta$, where $M'(y \rightarrow \hat{y}_2) = -M'(y \rightarrow \hat{y}_1) = 1$ and $M'(y' \rightarrow \hat{y}') = 0$ for all other $y', \hat{y}'$. Intuitively this corresponds to moving mass $M_{y \rightarrow \hat{y}_1}$ to / from $M_{y \rightarrow \hat{y}_2}$, in a way that does not violate any of the constraints. But $M$ is a vertex so it cannot be the convex combination of two feasible points.

But moreover, $S_M$ for an optimal vertex solution has exactly one U per row and the rest L's and 0's. Suppose the row corresponding to $y$ has $S_M(y, \hat{y})$ either L or S and the rest of the entries either L or 0. From the observations above of the submatrix $M^{(k)}$, there exists a row $y'$ such that $S_M(y', \hat{y}) = $ U. Then, we have $\sum_{\hat{y}} M_{y \rightarrow \hat{y}} < \sum_{\hat{y}} M_{y' \rightarrow \hat{y}} = 1$, which contradicts feasibility. Similarly, if a row of $S_M$ contains a U and an S (or multiple U's), then we would have $\sum_{\hat{y}} M_{y \rightarrow \hat{y}} > \sum_{\hat{y}} M_{y' \rightarrow \hat{y}}$.

This allows us to define $\Psi : \mathcal{Y} \rightarrow \mathcal{O}$ to be the unique $\hat{y}$ such that $S_M(y, \hat{y}) = $ U. Note that $\Psi$ is non-decreasing from the observation earlier that $S_M$ cannot contain the $2 \times 2$ signature above. This completely characterizes $S_M$ as the form described by Claim 10(2a–2c), completing the proof of Theorem 8. $\qquad \square$

We use a continuity argument to show that Theorem 8 holds in the case where the loss function $\ell(\hat{y}, y)$ is decreasing / increasing instead of *strictly* decreasing / increasing.

**Corollary 13.** *Let $P$ be a distribution over $\mathbb{R}$ with finite support, $\mathcal{O}$ a finite subset of $\mathbb{R}$, and $\ell : \mathbb{R} \times \mathbb{R} \rightarrow \mathbb{R}_{\geq 0}$ is such that*

- *For all $y \in \mathbb{R}$, $\ell(\hat{y}, y)$ is decreasing in $\hat{y}$ when $\hat{y} \leq y$ and increasing in $\hat{y}$ when $\hat{y} \geq y$.*
- *For all $\hat{y} \in \mathbb{R}$, $\ell(\hat{y}, y)$ is decreasing in $y$ when $y \leq \hat{y}$ and increasing in $y$ when $y \geq \hat{y}$.*

*Then for all $\varepsilon > 0$, there exists $\Phi : \mathcal{Y} \rightarrow \mathcal{O}$ such that*

$$\mathcal{L}(\text{RR-on-Bins}_\varepsilon^\Phi; P) = \inf_{\mathcal{M}} \mathcal{L}(\mathcal{M}; P).$$

*where $\inf_{\mathcal{M}}$ is over all $\varepsilon$-DP mechanisms $\mathcal{M}$ with inputs in $\mathcal{Y}$ and outputs in $\mathcal{O}$.*

*Proof.* Let $V$ correspond to the set of solutions $M$ that are vertices of the LP. Let $W \subseteq V$ correspond to the set of vertex solutions that are RR-on-Bins. A rephrasing of Theorem 8 is that the minimum loss over mechanisms in $V$ is equal to the minimum loss over mechanisms in $W$.

For $\eta > 0$, define $\ell_\eta(\hat{y}, y) = \ell(\hat{y}, y) + \eta \cdot |y - \hat{y}|$. Note that $\ell_\eta$ satisfies the conditions of Theorem 8. For any fixed mechanism $M$, the loss is continuous at $\eta = 0$:

$$\lim_{\eta \rightarrow 0} \sum_{y \in \mathcal{Y}} p_y \left( \sum_{\hat{y} \in \mathcal{O}} M_{y \rightarrow \hat{y}} \cdot \ell_\eta(\hat{y}, y) \right) = \sum_{y \in \mathcal{Y}} p_y \left( \sum_{\hat{y} \in \mathcal{O}} M_{y \rightarrow \hat{y}} \cdot \ell(\hat{y}, y) \right).$$

The minimum of finitely many continuous functions is continuous. In particular

$$\lim_{\eta \rightarrow 0} \min_{M \in V} \sum_{y \in \mathcal{Y}} p_y \left( \sum_{\hat{y} \in \mathcal{O}} M_{y \rightarrow \hat{y}} \cdot \ell_\eta(\hat{y}, y) \right) = \min_{M \in V} \sum_{y \in \mathcal{Y}} p_y \left( \sum_{\hat{y} \in \mathcal{O}} M_{y \rightarrow \hat{y}} \cdot \ell(\hat{y}, y) \right),$$

and similarly

$$\lim_{\eta \rightarrow 0} \min_{M \in W} \sum_{y \in \mathcal{Y}} p_y \left( \sum_{\hat{y} \in \mathcal{O}} M_{y \rightarrow \hat{y}} \cdot \ell_\eta(\hat{y}, y) \right) = \min_{M \in W} \sum_{y \in \mathcal{Y}} p_y \left( \sum_{\hat{y} \in \mathcal{O}} M_{y \rightarrow \hat{y}} \cdot \ell(\hat{y}, y) \right),$$

but the LHS of both equations are equal by Theorem 8, so

$$\min_{M \in V} \sum_{y \in \mathcal{Y}} p_y \left( \sum_{\hat{y} \in \mathcal{O}} M_{y \rightarrow \hat{y}} \cdot \ell(\hat{y}, y) \right) = \min_{M \in W} \sum_{y \in \mathcal{Y}} p_y \left( \sum_{\hat{y} \in \mathcal{O}} M_{y \rightarrow \hat{y}} \cdot \ell(\hat{y}, y) \right),$$

completing our proof. $\qquad \square$

**Stage II: Finitely supported $P$ and $\hat{\mathcal{Y}} \subseteq \mathbb{R}$.** We now set out to prove Theorem 5. Let $y_{\min}$ and $y_{\max}$ denote the minimum and maximum values in $\mathcal{Y}$, respectively. We will first show that

$$\inf_{\hat{\mathcal{Y}} \subseteq \mathbb{R}} \inf_{\Phi: \mathcal{Y} \to \hat{\mathcal{Y}}} \mathcal{L}(\mathsf{RR\text{-}on\text{-}Bins}_\varepsilon^\Phi; P) = \inf_{\mathcal{M}} \mathcal{L}(\mathcal{M}; P), \tag{3}$$

where the infimum on the RHS is over all $\varepsilon$-DP mechanisms $\mathcal{M}$.

Since $\mathsf{RR\text{-}on\text{-}Bins}_\varepsilon^\Phi$ is an $\varepsilon$-DP mechanism, we have

$$\inf_{\mathcal{M}} \mathcal{L}(\mathcal{M}; P) \leq \inf_{\hat{\mathcal{Y}} \subseteq \mathbb{R}} \inf_{\Phi: \mathcal{Y} \to \hat{\mathcal{Y}}} \mathcal{L}(\mathsf{RR\text{-}on\text{-}Bins}_\varepsilon^\Phi; P). \tag{4}$$

To show the converse, let $\gamma > 0$ be any parameter. There must exist an $\varepsilon$-DP mechanism $\mathcal{M}' : \mathcal{Y} \to \mathbb{R}$ such that

$$\mathcal{L}(\mathcal{M}'; P) \leq \inf_{\mathcal{M}} \mathcal{L}(\mathcal{M}; P) + \gamma/2. \tag{5}$$

Let $\mathcal{O} \subseteq [y_{\min}, y_{\max}]$ be defined as follows:

- For each $y \in \mathcal{Y}$, let $a_y = \min_{\hat{y} \in [y_{\min}, y_{\max}]} \ell(\hat{y}, y)$ and $b_y = \max_{\hat{y} \in [y_{\min}, y_{\max}]} \ell(\hat{y}, y)$. Let $T := \lceil 4(b_y - a_y)/\gamma \rceil$.
- Let $o_{y,t}$ be the finite set containing the maximal and minimal element of $\{\hat{y} \mid \ell(\hat{y}, y) = a_y + \frac{t}{T}(b_y - a_y)\}$ if the set is non-empty. Otherwise, let it be the empty set.
- Let $\mathcal{O}_y := \bigcup_{t=0}^{T} o_{y,t}$.
- Finally, let $\mathcal{O} = \bigcup_{y \in \mathcal{Y}} \mathcal{O}_y$.

Naturally, $\mathcal{O}$ is finite. Finally, let $\mathcal{M}''$ be the mechanism that first runs $\mathcal{M}'$ to get $\hat{y}$ and then outputs the element in $\mathcal{O}$ closest to $\hat{y}$. By post-processing of DP, $\mathcal{M}''$ remains $\varepsilon$-DP. Furthermore, it is not hard to see that by the construction of $\mathcal{O}$ and by Assumption 4, we have

$$\mathcal{L}(\mathcal{M}''; P) \leq \mathcal{L}(\mathcal{M}'; P) + \gamma/2. \tag{6}$$

Finally, since $\mathrm{range}(\mathcal{M}'') = \mathcal{O}$ is finite, the proof in Stage I implies that

$$\inf_{\hat{\mathcal{Y}} \subseteq \mathcal{O}} \inf_{\Phi: \mathcal{Y} \to \hat{\mathcal{Y}}} \mathcal{L}(\mathsf{RR\text{-}on\text{-}Bins}_\varepsilon^\Phi; P) \leq \mathcal{L}(\mathcal{M}''; P). \tag{7}$$

Combining (5), (6), and (7), we can conclude that

$$\inf_{\hat{\mathcal{Y}} \subseteq \mathbb{R}} \inf_{\Phi: \mathcal{Y} \to \hat{\mathcal{Y}}} \mathcal{L}(\mathsf{RR\text{-}on\text{-}Bins}_\varepsilon^\Phi; P) \leq \inf_{\mathcal{M}} \mathcal{L}(\mathcal{M}; P) + \gamma. \tag{8}$$

Since (8) holds for any $\gamma > 0$, combining with (4), we can conclude that (3) holds.

Next, we will show that there exists $\hat{\mathcal{Y}}^*$ and $\Phi^* : \mathcal{Y} \to \hat{\mathcal{Y}}^*$ such that

$$\mathcal{L}(\mathsf{RR\text{-}on\text{-}Bins}_\varepsilon^{\Phi^*}; P) = \inf_{\hat{\mathcal{Y}} \subseteq \mathbb{R}} \inf_{\Phi: \mathcal{Y} \to \hat{\mathcal{Y}}} \mathcal{L}(\mathsf{RR\text{-}on\text{-}Bins}_\varepsilon^\Phi; P). \tag{9}$$

Combining this with (3) completes the proof.

For any $\Phi : \mathcal{Y} \to \mathbb{R}$, let $\mathcal{P}_\Phi$ denote the partition on $\mathcal{Y}$ induced by $\Phi^{-1}$. Note that the RHS of (9) can be written as

$$\inf_{\hat{\mathcal{Y}} \subseteq \mathbb{R}} \inf_{\Phi: \mathcal{Y} \to \hat{\mathcal{Y}}} \mathcal{L}(\mathsf{RR\text{-}on\text{-}Bins}_\varepsilon^\Phi; P) = \min_{\mathcal{P}} \inf_{\substack{\Phi: \mathcal{Y} \to \hat{\mathcal{Y}} \\ \hat{\mathcal{Y}} \subseteq \mathbb{R} \\ \mathcal{P}_\Phi = \mathcal{P}}} \mathcal{L}(\mathsf{RR\text{-}on\text{-}Bins}_\varepsilon^\Phi; P), \tag{10}$$

where the minimum is over all partitions of $\mathcal{P}$.

For a fixed partition $\mathcal{P} = \mathcal{P}_\Phi$ of $\mathcal{Y}$, $\mathcal{L}(\mathsf{RR\text{-}on\text{-}Bins}_\varepsilon^\Phi; P)$ can simply be written as

$$\sum_{S_i \in \mathcal{P}} \left( \sum_{y \in \mathcal{Y}} p_y \frac{e^{\varepsilon \mathbb{1}[y \in S_i]}}{e^\varepsilon + |\hat{\mathcal{Y}}| - 1} \ell(\hat{y}_i, y) \right),$$

where $\hat{y}_i$ is the output corresponding to the part $S_i$ in the partition.

Notice that the function $\hat{y}_i \rightarrow \left( \sum_{y \in \mathcal{Y}} p_y \frac{e^{\varepsilon \mathbb{1}[y \in S_i]}}{e^{\varepsilon} + |\hat{\mathcal{Y}}| - 1} \ell(\hat{y}_i, y) \right)$ is continuous. Furthermore, it is obvious that it increases once it becomes further from $[y_{\min}, y_{\max}]$. Thus, the minimum must be achieved at some point $\hat{y}_i^* \in [y_{\min}, y_{\max}]$. As a result, by defining $\Phi_{\mathcal{P}}^*$ such that $\Phi_{\mathcal{P}}^*(S_i) = \hat{y}_i^*$, this also achieves the minimum for $\inf_{\substack{\Phi : \mathcal{Y} \rightarrow \hat{y} \\ \hat{\mathcal{Y}} \subseteq \mathbb{R} \\ \mathcal{P}_{\Phi} = \mathcal{P}}} \mathcal{L}(\mathsf{RR\text{-}on\text{-}Bins}_{\varepsilon}^{\Phi}; P)$. Therefore, plugging this back into

(10), we can conclude that the minimum of $\inf_{\hat{\mathcal{Y}} \subseteq \mathbb{R}} \inf_{\Phi : \mathcal{Y} \rightarrow \hat{y}} \mathcal{L}(\mathsf{RR\text{-}on\text{-}Bins}_{\varepsilon}^{\Phi}; P)$ must be achieved by some $\hat{\mathcal{Y}}^*, \Phi^*$. This completes our proof of Theorem 5.

### A.1 EXTENSION TO ARBITRARY (INFINITELY-SUPPORTED) $P$ AND $\hat{\mathcal{Y}} \subseteq \mathbb{R}$.

We show that Theorem 5 can be extended to hold even for infinitely-supported distributions $P$ over a bounded interval, but under an additional assumption that the loss $\ell$ is Lipschitz over the said interval.

**Assumption 14.** *For a specified bounded interval $[y_{\min}, y_{\max}]$, loss function $\ell : [y_{\min}, y_{\max}] \times [y_{\min}, y_{\max}] \rightarrow \mathbb{R}_{\geq 0}$ is such that both $\ell(\cdot, y)$ and $\ell(\hat{y}, \cdot)$ are $L$-Lipschitz.*

**Theorem 15.** *For all $y_{\min}, y_{\max} \in \mathbb{R}$, loss functions $\ell : \mathbb{R} \times \mathbb{R} \rightarrow \mathbb{R}_{\geq 0}$ satisfying Assumptions 4 and 14, and all distributions $P$ over $\mathcal{Y} \subseteq [y_{\min}, y_{\max}]$, there is a finite output set $\hat{\mathcal{Y}} \subseteq \mathbb{R}$ and a non-decreasing map $\Phi : \mathcal{Y} \rightarrow \hat{y}$ such that*
$$\mathcal{L}(\mathsf{RR\text{-}on\text{-}Bins}_{\varepsilon}^{\Phi}; P) = \inf_{\mathcal{M}} \mathcal{L}(\mathcal{M}; P),$$
*where the infimum is over all $\varepsilon$-DP mechanisms $\mathcal{M}$.*

We break down the proof into a series of lemmas. First we show that the infimum is reached by the infimum of RR-on-Bins mechanisms:

**Lemma 16.** *Under the assumptions of Theorem 15, it holds that*
$$\inf_{\substack{\hat{\mathcal{Y}} \subseteq \mathbb{R}, \\ \Phi : \mathcal{Y} \rightarrow \hat{y}}} \mathcal{L}(\mathsf{RR\text{-}on\text{-}Bins}_{\varepsilon}^{\Phi}; P) = \inf_{\mathcal{M}} \mathcal{L}(\mathcal{M}; P).$$

*Proof.* To prove the lemma, it suffices to show that for any $g \in \mathbb{N}$, there exists a choice of $\hat{\mathcal{Y}} \subseteq \mathbb{R}$ and $\Phi : \mathcal{Y} \rightarrow \hat{y}$ such that $\mathcal{L}(\mathsf{RR\text{-}on\text{-}Bins}_{\varepsilon}^{\Phi}; P) \leq \inf_{\mathcal{M}} \mathcal{L}(\mathcal{M}; P) + \gamma$ where $\gamma = L \cdot \frac{y_{\max} - y_{\min}}{g}$.

Consider a discretization of $\mathcal{Y}$ given as
$$\tilde{\mathcal{Y}} := \left\{ y_{\min} + \frac{i\gamma}{L} : 0 \leq i \leq g \right\}.$$

Let $\rho : \mathcal{Y} \rightarrow \tilde{\mathcal{Y}}$ be the map that rounds any element of $\mathcal{Y}$ to the closes element of $\tilde{\mathcal{Y}}$. Note that $|y - \rho(y)| \leq \gamma/(2L)$ for all $y \in \mathcal{Y}$. Consider the distribution $\tilde{P}$ over $\tilde{\mathcal{Y}}$ given by the "rounding process", which samples $y \sim P$ and returns $\rho(y)$. Note that there is a natural coupling between $P$ and $\tilde{P}$ such that $|y - \hat{y}| \leq \gamma/(2L)$ holds with probability 1. From Theorem 5, we have that there exists a finite $\hat{\mathcal{Y}} \subseteq \mathbb{R}$ and non-decreasing map $\tilde{\Phi} : \tilde{\mathcal{Y}} \rightarrow \hat{y}$ such that
$$\mathcal{L}(\mathsf{RR\text{-}on\text{-}Bins}_{\varepsilon}^{\tilde{\Phi}}; \tilde{P}) = \inf_{\tilde{\mathcal{M}}} \mathcal{L}(\tilde{\mathcal{M}}; \tilde{P}), \tag{11}$$

where $\tilde{\mathcal{M}}$ is an $\varepsilon$-DP mechanism mapping $\tilde{\mathcal{Y}}$ to $\hat{y}$. We can extend $\tilde{\Phi}$ to $\Phi : \mathcal{Y} \rightarrow \hat{y}$ given as $\Phi(y) := \tilde{\Phi}(\rho(y))$. It is easy to see that since $\tilde{\Phi}$ is non-decreasing, $\Phi$ is also non-decreasing. From Assumption 14, it follows that
$$\mathcal{L}(\mathsf{RR\text{-}on\text{-}Bins}_{\varepsilon}^{\Phi}; P) = \mathop{\mathbb{E}}_{y \sim P} \ell(\mathsf{RR\text{-}on\text{-}Bins}_{\varepsilon}^{\Phi}(y), y)$$
$$= \mathop{\mathbb{E}}_{y \sim P} \ell(\mathsf{RR\text{-}on\text{-}Bins}_{\varepsilon}^{\tilde{\Phi}}(\rho(y)), y)$$
$$\leq \mathop{\mathbb{E}}_{y \sim P} \ell(\mathsf{RR\text{-}on\text{-}Bins}_{\varepsilon}^{\tilde{\Phi}}(\rho(y)), \rho(y)) + \gamma/2 \quad \text{(from Assumption 14)}$$
$$= \mathcal{L}(\mathsf{RR\text{-}on\text{-}Bins}_{\varepsilon}^{\tilde{\Phi}}; \tilde{P}) + \gamma/2. \tag{12}$$

Similarly, for any $\varepsilon$-DP mechanism $\mathcal{M}$ mapping $\mathcal{Y}$ to $\hat{\mathcal{Y}}$, we can construct an $\varepsilon$-DP mechanism $\tilde{\mathcal{M}}$ mapping $\tilde{\mathcal{Y}}$ to $\hat{\mathcal{Y}}$ where $\tilde{\mathcal{M}}(\tilde{y})$ is sampled as $\mathcal{M}(y)$ for $y \sim P|_{\rho(y)=\tilde{y}}$. Note that sampling $\tilde{y} \sim \tilde{P}$ and returning $y \sim P|_{\rho(y)=\tilde{y}}$ is equivalent to sampling $y \sim P$. Thus, we have

$$
\begin{aligned}
\mathcal{L}(\tilde{\mathcal{M}}; \tilde{P}) &= \underset{\tilde{y} \sim \tilde{P}}{\mathbb{E}} \ell(\tilde{M}(\tilde{y}), \tilde{y}) \\
&= \underset{y \sim P}{\mathbb{E}} \ell(M(y), \rho(y)) \\
&\leq \underset{y \sim P}{\mathbb{E}} \ell(M(y), y) + \gamma/2 \qquad \text{(from Assumption 14)} \\
&= \mathcal{L}(\mathcal{M}; P) + \gamma/2.
\end{aligned} \tag{13}
$$

Thus, combining (11), (12), and (13), we get

$$
\begin{aligned}
\mathcal{L}(\mathsf{RR\text{-}on\text{-}Bins}_\varepsilon^\Phi; P) &\leq \mathcal{L}(\mathsf{RR\text{-}on\text{-}Bins}_\varepsilon^{\tilde{\Phi}}; \tilde{P}) + \gamma/2 \\
&= \inf_{\tilde{\mathcal{M}}} \mathcal{L}(\tilde{\mathcal{M}}; \tilde{P}) + \gamma/2 \\
&\leq \inf_{\mathcal{M}} \mathcal{L}(\mathcal{M}; P) + \gamma. \qquad \square
\end{aligned}
$$

Next we show that the infimum of RR-on-Bins is reached by considering the RR-on-Bins with finitely many bins. Towards this end, let $\mathsf{RR\text{-}on\text{-}Bins}_\varepsilon^n$ denote the set of all $\mathsf{RR\text{-}on\text{-}Bins}_\varepsilon^\Phi$ mechanisms where $\Phi : \mathcal{Y} \to \hat{\mathcal{Y}}$ with $|\hat{\mathcal{Y}}| = n$.

**Lemma 17.** *Suppose $\ell(\hat{y}, y)$ is integrable over $y$ for any $\hat{y}$. Then for all $\varepsilon > 0$, it holds that*

$$
\lim_{n \to \infty} \inf_{\mathcal{M} \in \mathsf{RR\text{-}on\text{-}Bins}_\varepsilon^n} \mathcal{L}(\mathcal{M}; P) \geq \inf_{\mathcal{M} \in \mathsf{RR\text{-}on\text{-}Bins}_\varepsilon^1} \mathcal{L}(\mathcal{M}; P).
$$

*Remark:* Interestingly, Lemma 17 does not require any assumption about the distribution $P$ or $\ell$ aside from integrability. For example, $P$ can be an unbounded distribution.

*Proof.* Let

$$
\alpha := \inf_{\mathcal{M} \in \mathsf{RR\text{-}on\text{-}Bins}_\varepsilon^1} \mathcal{L}(\mathcal{M}; P) = \inf_{\hat{y} \in \mathbb{R}} \int \ell(\hat{y}, y) dP(y).
$$

This is precisely $\inf_{\mathcal{M} \in \mathsf{RR\text{-}on\text{-}Bins}_\varepsilon^1} \mathcal{L}(\mathcal{M}; P)$, where the $\hat{y}$ is selecting the single bin to output to. Note that $\alpha$ does not depend on $\varepsilon$.

For any $n > 1$, consider $\mathsf{RR\text{-}on\text{-}Bins}_\varepsilon^\Phi$, where $\Phi : \mathcal{Y} \to \hat{\mathcal{Y}}$ for $\hat{\mathcal{Y}} = \{\hat{y}_1, \ldots, \hat{y}_n\}$. Then we can bound the loss from below:

$$
\begin{aligned}
\mathcal{L}(\mathsf{RR\text{-}on\text{-}Bins}_\varepsilon^\Phi; P) &= \int \left( \left( \sum_{i=1}^n \frac{1}{e^\varepsilon + n - 1} \ell(\hat{y}_i, y) \right) + \frac{e^\varepsilon - 1}{e^\varepsilon + n - 1} \ell(\Phi(y), y) \right) dP(y) \\
&\geq \int \left( \sum_{i=1}^n \frac{1}{e^\varepsilon + n - 1} \ell(\hat{y}_i, y) \right) dP(y) \\
&= \sum_{i=1}^n \int \frac{1}{e^\varepsilon + n - 1} \ell(\hat{y}_i, y) dP(y) \\
&\geq \sum_{i=1}^n \frac{1}{e^\varepsilon + n - 1} \cdot \alpha \\
&= \frac{n}{e^\varepsilon + n - 1} \cdot \alpha.
\end{aligned}
$$

In particular, $\inf_{\mathcal{M} \in \mathsf{RR\text{-}on\text{-}Bins}_\varepsilon^n} \mathcal{L}(\mathcal{M}; P) \geq \frac{n}{e^\varepsilon + n - 1} \alpha$ which implies that

$$
\lim_{n \to \infty} \inf_{\mathcal{M} \in \mathsf{RR\text{-}on\text{-}Bins}_\varepsilon^n} \mathcal{L}(\mathcal{M}; P) \geq \alpha. \qquad \square
$$

**Corollary 18.** *Suppose $\ell(\hat{y}, y)$ is integrable over $y$ for any $\hat{y}$. Then for all $\varepsilon > 0$, there exists $n \geq 1$ such that*

$$\inf_{\mathcal{M} \in \mathsf{RR\text{-}on\text{-}Bins}_\varepsilon^n} \mathcal{L}(\mathcal{M}; P) = \inf_n \inf_{\mathcal{M} \in \mathsf{RR\text{-}on\text{-}Bins}_\varepsilon^n} \mathcal{L}(\mathcal{M}; P).$$

*Proof.* For any $n$, let $\alpha_n := \inf_{\mathcal{M} \in \mathsf{RR\text{-}on\text{-}Bins}_\varepsilon^n} \mathcal{L}(\mathcal{M}; P)$. From Lemma 17, we have that $\lim_{n \to \infty} \alpha_n \geq \alpha_1$. If $\inf_n \alpha_n = \alpha_1$, then the corollary is true for $n = 1$. Else, if $\inf_n \alpha_n = \alpha' < \alpha_1$, then there exists $n_0 > 0$ such that for all $n > n_0$, it holds that $\alpha_n > (\alpha_1 + \alpha')/2$. Thus, $\inf_n \alpha_n = \min_{n \leq n_0} \alpha_n$ which implies that the infimum is realized for some finite $n$. $\square$

Finally, we show that the infimum over $\mathsf{RR\text{-}on\text{-}Bins}_\varepsilon^n$ is also achievable, completing the proof of Theorem 15.

**Lemma 19.** *Suppose $P$ is bounded within the interval $\mathcal{Y} = [y_{\min}, y_{\max}]$. Suppose $\ell : \mathbb{R} \times \mathbb{R} \to \mathbb{R}_{\geq 0}$ satisfies Assumptions 4 and 14, then for any $n > 0$, there is an output set $\hat{\mathcal{Y}} \subseteq \mathbb{R}$ with $|\hat{\mathcal{Y}}| = n$ and a non-decreasing map $\Phi : \mathcal{Y} \to \hat{\mathcal{Y}}$ such that*

$$\mathcal{L}(\mathsf{RR\text{-}on\text{-}Bins}_\varepsilon^\Phi; P) = \inf_{\mathcal{M} \in \mathsf{RR\text{-}on\text{-}Bins}_\varepsilon^n} \mathcal{L}(\mathcal{M}; P).$$

*Proof.* Because of the increasing / decreasing nature of $\ell$, we can restrict the output of $\Phi$ to be in the range $[y_{\min}, y_{\max}]$. Given an output range $\{\hat{y}_1, \ldots, \hat{y}_n\}$, the $\mathsf{RR\text{-}on\text{-}Bins}_\varepsilon^\Phi$ mechanism that minimizes $\mathcal{L}(\mathsf{RR\text{-}on\text{-}Bins}_\varepsilon^\Phi; P)$ satisfies

$$\Phi(y) = \arg\min_{\hat{y}_i} \ell(\hat{y}_i, y).$$

So we can consider $\mathcal{L}(\mathsf{RR\text{-}on\text{-}Bins}_\varepsilon^\Phi; P)$ as a function of $(\hat{y}_1, \ldots, \hat{y}_n) \in [y_{\min}, y_{\max}]^n$.

We claim that $\mathcal{L}(\mathsf{RR\text{-}on\text{-}Bins}_\varepsilon^\Phi; P)$ is continuous with respect to these inputs $\{\hat{y}_1, \ldots, \hat{y}_n\}$. Indeed by the Lipschitz continuity, a change in $[\hat{y}_1, \ldots, \hat{y}_n]$ to $[\hat{y}_1', \ldots, \hat{y}_n']$ where $|\hat{y}_i - \hat{y}_i'| < \delta$ for some $\delta > 0$ results in a change in $\mathcal{L}(\mathsf{RR\text{-}on\text{-}Bins}_\varepsilon^\Phi; P)$ bounded by $L\delta$. Therefore $\mathcal{L}(\mathsf{RR\text{-}on\text{-}Bins}_\varepsilon^\Phi; P)$ is a continuous function of $(\hat{y}_1, \ldots, \hat{y}_n)$ over a compact set $[y_{\min}, y_{\max}]^n$ and hence it attains its infimum. This infimum defines $\Phi$ that satisfies the lemma. $\square$

## B ANALYSIS OF DYNAMIC PROGRAMMING ALGORITHM FOR FINDING OPTIMAL RR-on-Bins

Below we give correctness and running time analysis of Algorithm 2.

**Correctness Proof.** We will prove the following statement by strong induction on $i$:

$$A[i][j] = \min_{\substack{\Phi : \{y^1, \ldots, y^i\} \to \mathbb{R} \\ |\mathcal{P}_\Phi| = j}} \sum_{S \in \mathcal{P}_\Phi} \sum_{y \in \mathcal{Y}} p_y \cdot e^{\mathbb{1}[y \in S] \cdot \varepsilon} \cdot \ell(\Phi(S), y), \tag{14}$$

where the minimum is across all $\Phi : \{y^1, \ldots, y^i\} \to \mathbb{R}$ such that, for each $\hat{y} \in \mathrm{range}(\Phi)$, $\Phi^{-1}(\hat{y})$ is an interval and that $|\mathcal{P}_\Phi| = j$ where the $j$ intervals are $S_1, \ldots, S_j$ (in increasing order).

Before we prove the above statement, note that it implies that $A[n][d] = (d - 1 + e^\varepsilon) \cdot \min_{\hat{\mathcal{Y}}, \Phi : \mathcal{Y} \to \hat{\mathcal{Y}}, |\mathcal{P}_\Phi| = d} \mathcal{L}(\mathsf{RR\text{-}on\text{-}Bins}_\varepsilon^\Phi; P)$. Thus, the last line of the algorithm ensures that we output the optimal RR-on-Bins as desired.

We will now prove (14) via strong induction on $i$.

- **Base Case.** For $i = 0$ (and thus $j = 0$), the statement is obviously true.

- **Inductive Step.** Now, suppose that the statement is true for all $i = 0, \ldots, t-1$ for some $t \in \mathbb{N}$. We will show that it is also true for $i = t$. To see this, we may rewrite the RHS term (for $i = t$) as

$$\min_{\substack{\Phi:\{y^1,\ldots,y^t\}\to\mathbb{R} \\ |\mathcal{P}_\Phi|=j}} \sum_{S\in\mathcal{P}_\Phi} \sum_{y\in\mathcal{Y}} p_y \cdot e^{\mathbb{1}[y\in S]\cdot\varepsilon} \cdot \ell(\Phi(S), y)$$

$$= \min_{0\leq r<t} \min_{\substack{\Phi:\{y^1,\ldots,y^t\}\to\mathbb{R} \\ |\mathcal{P}_\Phi|=j \\ \{y^{r+1},\ldots,y^t\}\in\mathcal{P}_\Phi}} \sum_{S\in\mathcal{P}_\Phi} \sum_{y\in\mathcal{Y}} p_y \cdot e^{\mathbb{1}[y\in S]\cdot\varepsilon} \cdot \ell(\Phi(S), y)$$

$$= \min_{0\leq r<t} \left( \min_{\substack{\Phi:\{y^1,\ldots,y^r\}\to\mathbb{R} \\ |\mathcal{P}_\Phi|=j-1}} \sum_{S\in\mathcal{P}_\Phi} \sum_{y\in\mathcal{Y}} p_y \cdot e^{\mathbb{1}[y\in S]\cdot\varepsilon} \cdot \ell(\Phi(S), y) \right.$$

$$\left. + \min_{\hat{y}=\Phi(\{y^{r+1},\ldots,y^t\})} \sum_{y\in\mathcal{Y}} p_y \cdot e^{\mathbb{1}[y\in[y^{r+1},y^t]]\cdot\varepsilon} \cdot \ell(\hat{y}, y) \right)$$

$$= \min_{0\leq r<t} A[r][j-1] + L[r+1][t].$$

where the third inequality follows from the inductive hypothesis and the definition of $L[r+1][t]$. The last expression is exactly how $A[t][j]$ is computed in our algorithm. Thus, (14) holds for $i = t$.

**Running Time Analysis.** It is clear that, apart from the computation of $L[r+1][i]$, the remainder of the algorithm runs in time $O(k^2)$. Therefore, the total running time is $O(k^2 \cdot T)$ where $T$ denotes the running time for solving the problem $\min_{\hat{y}\in\mathbb{R}} \sum_{y\in\mathcal{Y}} p_y \cdot e^{\mathbb{1}[y\in[y^r,y^i]]\cdot\varepsilon} \cdot \ell(\hat{y}, y)$. When the loss function $\ell$ is convex, this problem is a univariate convex optimization problem and can be solved in polynomial time. We can even speed this up further. In fact, for the three main losses we consider (squared loss, Poisson log loss, and absolute-value loss) this problem can be solved in amortized constant time, as detailed below. Therefore, for these three losses, the total running time of the dynamic programming algorithm is only $O(k^2)$.

- **Squared Loss.**
  In this case, the minimizer is simply

$$\arg\min_{\hat{y}\in\mathbb{R}} \sum_{y\in\mathcal{Y}} p_y \cdot e^{\mathbb{1}[y\in[y^r,y^i]]\cdot\varepsilon} \cdot \ell_{\mathrm{sq}}(\hat{y}, y) = \frac{\sum_{y\in\mathcal{Y}} p_y \cdot e^{\mathbb{1}[y\in[y^r,y^i]]\cdot\varepsilon} y}{\sum_{y\in\mathcal{Y}} p_y \cdot e^{\mathbb{1}[y\in[y^r,y^i]]\cdot\varepsilon}} =: \hat{y}^*_{r,i}.$$

  Therefore, to compute $\hat{y}^*_{r,i}$, it suffices to keep the values $\sum_{y\in\mathcal{Y}} p_y \cdot e^{\mathbb{1}[y\in[y^r,y^i]]\cdot\varepsilon} y$ and $\sum_{y\in\mathcal{Y}} p_y \cdot e^{\mathbb{1}[y\in[y^r,y^i]]\cdot\varepsilon}$. We can start with $r = i$ and as we increase $i$, these quantities can be updated in constant time.
  Note also that

$$\min_{\hat{y}\in\mathbb{R}} \sum_{y\in\mathcal{Y}} p_y \cdot e^{\mathbb{1}[y\in[y^r,y^i]]\cdot\varepsilon} \cdot \ell_{\mathrm{sq}}(\hat{y}, y)$$

$$= \sum_{y\in\mathcal{Y}} p_y \cdot e^{\mathbb{1}[y\in[y^r,y^i]]\cdot\varepsilon} \cdot y^2 - 2 \left( \sum_{y\in\mathcal{Y}} p_y \cdot e^{\mathbb{1}[y\in[y^r,y^i]]\cdot\varepsilon} \cdot y \right) \hat{y}^*_{r,i} + (\hat{y}^*_{r,i})^2.$$

  Therefore, to compute the minimum value, we additionally keep $\sum_{y\in\mathcal{Y}} p_y \cdot e^{\mathbb{1}[y\in[y^r,y^i]]\cdot\varepsilon} y^2$, which again can be updated in constant time for each $i$.

- **Poisson Log Loss.**
  The minimizer is exactly the same as in the squared loss, i.e.,

$$\arg\min_{\hat{y}\in\mathbb{R}} \sum_{y\in\mathcal{Y}} p_y \cdot e^{\mathbb{1}[y\in[y^r,y^i]]\cdot\varepsilon} \cdot \ell_{\mathrm{Poi}}(\hat{y}, y) = \frac{\sum_{y\in\mathcal{Y}} p_y \cdot e^{\mathbb{1}[y\in[y^r,y^i]]\cdot\varepsilon} y}{\sum_{y\in\mathcal{Y}} p_y \cdot e^{\mathbb{1}[y\in[y^r,y^i]]\cdot\varepsilon}} =: \hat{y}^*_{r,i}.$$

Therefore, using the same method as above, we can compute $\hat{y}^*$ in amortized constant time. The minimum is the simply

$$\min_{\hat{y} \in \mathbb{R}} \sum_{y \in \mathcal{Y}} p_y \cdot e^{\mathbb{1}[y \in [y^r, y^i]] \cdot \varepsilon} \cdot \ell_{\text{Poi}}(\hat{y}, y)$$

$$= \left( \sum_{y \in \mathcal{Y}} p_y \cdot e^{\mathbb{1}[y \in [y^r, y^i]] \cdot \varepsilon} \right) \hat{y}_{r,i}^* - \left( \sum_{y \in \mathcal{Y}} p_y \cdot e^{\mathbb{1}[y \in [y^r, y^i]] \cdot \varepsilon} y \right) \log(\hat{y}_{r,i}^*),$$

so this can also be computed in constant time with the quantities that we have recorded.

- **Absolute-Value Loss.**
  To describe the minimizer, we define a weighted version of median. Let $\{(w_1, a_1), \ldots, (w_t, a_t)\}$ be a set of $t$ tuples such that $w_1, \ldots, w_t \in \mathbb{R}_{\geq 0}$ and $a_1, \ldots, a_t \in \mathbb{R}$ with $a_1 \leq \cdots \leq a_t$. The *weighted median* of $\{(w_1, a_1), \ldots, (w_t, a_t)\}$, denoted by $\text{wmed}(\{(w_1, a_1), \ldots, (w_t, a_t)\})$ is equal to the minimum value $a^*$ such that $\sum_{\substack{j \in [t] \\ a_j \leq a^*}} w_j \geq (\sum_{j \in [t]} w_j)/2$. It is not hard to see that

$$\arg\min_{\hat{y} \in \mathbb{R}} \sum_{y \in \mathcal{Y}} p_y \cdot e^{\mathbb{1}[y \in [y^r, y^i]] \cdot \varepsilon} \cdot \ell_{\text{abs}}(\hat{y}, y) = \text{wmed}\left( \left\{ \left( p_y \cdot e^{\mathbb{1}[y \in [y^r, y^i]] \cdot \varepsilon}, y \right) \right\}_{y \in \mathcal{Y}} \right) =: \hat{y}_{r,i}^*.$$

Notice also that we must have $\hat{y}_{r,i}^* \in \mathcal{Y}$. For a fixed $r$ (and varying $i$), the algorithm is now as follows: first compute $\hat{y}_{r,r}^*$, and also compute

$$w_{\text{lo}} := \sum_{\substack{y \in \mathcal{Y} \\ y \leq \hat{y}_{r,r}^*}} p_y \cdot e^{\mathbb{1}[y \in [y^r, y^i]] \cdot \varepsilon},$$

and

$$w_{\text{hi}} := \sum_{\substack{y \in \mathcal{Y} \\ y > \hat{y}_{r,r}^*}} p_y \cdot e^{\mathbb{1}[y \in [y^r, y^i]] \cdot \varepsilon}.$$

For $i = r + 1, \ldots, k$, initialize $\hat{y}_{r,i}^* = \hat{y}_{r,i-1}^*$ and update $w_{\text{lo}}$ or $w_{\text{hi}}$ (corresponding to the weight change from $p_{y_i}$ to $e^\varepsilon p_{y_i}$ of $y_i$). We then perform updates to reach the correct value of $\hat{y}_{r,i}^*$ That is, if $w_{\text{lo}} < w_{\text{hi}}$, then move to the next larger value in $\mathcal{Y}$; if $w_{\text{lo}} - p_{\hat{y}_{r,i}^*} \cdot e^{\mathbb{1}[\hat{y}_{r,i}^* \in [y^r, y^i]] \cdot \varepsilon} \geq w_{\text{hi}}$, then move to the next smaller value in $\mathcal{Y}$. Otherwise, stop and keep the current $\hat{y}_{r,i}^*$.

To understand the running time of this subroutine, note that the initial running time for computing $\hat{y}_{r,r}^*$ and $w_{\text{lo}}, w_{\text{hi}}$ is $O(k)$. Furthermore, each update for $\hat{y}_{r,i}^*$ takes $O(1)$ time. Finally, observe that if we ever move $\hat{y}_{r,i}^*$ to a larger value, it must be that $y_i > \hat{y}_{r,i}^*$. After this $i$, $\hat{y}_{r,i}^*$ will never decrease again. As a result, in total across all $i = r + 1, \ldots, k$, the total number of updates can be at most $2k$. Thus, the total running time for a fixed $r$ is $O(k)$. Summing up across $r \in [k]$, we can conclude that the total running time of the entire dynamic programming algorithm is $O(k^2)$.

## C    RR-on-Bins WITH APPROXIMATE PRIOR

Recall from Section 3.2 that, when the prior is unknown, we split the budget into $\varepsilon_1, \varepsilon_2$, use the $\varepsilon_1$-DP Laplace mechanism to approximate the prior and then run the RR-on-Bins with privacy parameter $\varepsilon_2$. The remainder of this section gives omitted details and proofs from Section 3.2. Throughout this section, we use $k$ to denote $|\mathcal{Y}|$ (the size of the input label set).

### C.1    LAPLACE MECHANISM AND ITS GUARANTEES

We start by recalling the Laplace mechanism for estimating distribution. Recall that the Laplace distribution with scale parameter $b$, denoted by $\text{Lap}(b)$, is the distribution supported on $\mathbb{R}$ whose probability density function is $\frac{1}{2b} \exp(-|x|/b)$. The Laplace mechanism is presented in Algorithm 4.

It is well-known (e.g., Dwork et al. (2006b)) that this mechanism satisfies $\varepsilon$-DP. Its utility guarantee, which we will use in the analysis of LabelRandomizer, is also well-known:

---

**Algorithm 4** Laplace Mechanism for Estimating Probability Distribution $\mathcal{M}_\varepsilon^{\mathrm{Lap}}$.

---

**Parameters:** Privacy parameter $\varepsilon \geq 0$.
**Input:** Labels $y_1, \ldots, y_n \in \mathcal{Y}$.
**Output:** A probability distribution $P'$ over $\mathcal{Y}$.

**for** $y \in \mathcal{Y}$ **do**
    $h_y \leftarrow$ number of $i$ such that $y_i = y$
    $h'_y \leftarrow \max\{h_y + \mathrm{Lap}(2/\varepsilon), 0\}$
**return** Distribution $P'$ over $\mathcal{Y}$ such that $p'_y = \frac{h'_y}{\sum_{y \in \mathcal{Y}} h'_y}$

---

**Theorem 20** (e.g., Diakonikolas et al. (2015)). *For any distribution $P$ on $\mathcal{Y}$, $n \in \mathbb{N}$, and $\varepsilon > 0$, we have*

$$\mathop{\mathbb{E}}_{\substack{y_1, \ldots, y_n \sim P \\ P' \sim \mathcal{M}_\varepsilon^{\mathrm{Lap}}(y_1, \ldots, y_n)}} [\|P' - P\|_1] \leq O\left(\sqrt{\frac{k}{n}} + \frac{k}{\varepsilon n}\right).$$

## C.2    PROOF OF THEOREM 6

**Theorem 6.** $\mathsf{LabelRandomizer}_{\varepsilon_1, \varepsilon_2}$ *is* $(\varepsilon_1 + \varepsilon_2)$-*DP.*

*Proof of Theorem 6.* The Laplace mechanism is $\varepsilon_1$-DP; by post-processing property of DP, $\Phi'$ is also $\varepsilon_1$-DP. For fixed $\Phi'$, since $\mathsf{RR\text{-}on\text{-}Bins}_{\varepsilon_2}^{\Phi'}$ is $\varepsilon_2$-DP and it is applied only once on each label, the parallel composition theorem ensures that $(\hat{y}_1, \ldots, \hat{y}_n)$ is $\varepsilon_2$-DP. Finally, applying the basic composition theorem, we can conclude that the entire algorithm is $(\varepsilon_1 + \varepsilon_2)$-DP. $\square$

## C.3    PROOF OF THEOREM 7

**Theorem 7.** *Let $\ell : \mathbb{R} \times \mathbb{R} \to \mathbb{R}_{\geq 0}$ be any loss function satisfying Assumption 4. Furthermore, assume that $\ell(\hat{y}, y) \leq B$ for some parameter $B$ for all $y, \hat{y} \in \mathcal{Y}$. For any distribution $P$ on $\mathcal{Y}$, $\varepsilon > 0$, and any sufficiently large $n \in \mathbb{N}$, there is a choice of $\varepsilon_1, \varepsilon_2 > 0$ such that $\varepsilon_1 + \varepsilon_2 = \varepsilon$ and*

$$\mathop{\mathbb{E}}_{\substack{y_1, \ldots, y_n \sim P \\ P', \Phi', \hat{\mathcal{Y}}'}} [\mathcal{L}(\mathsf{RR\text{-}on\text{-}Bins}_{\varepsilon_2}^{\Phi'}; P)] - \inf_{\mathcal{M}} \mathcal{L}(\mathcal{M}; P) \leq O\left(B \cdot \sqrt{|\mathcal{Y}|/n}\right),$$

*where $P', \Phi', \hat{\mathcal{Y}}'$ are as in $\mathsf{LabelRandomizer}_{\varepsilon_1, \varepsilon_2}$ and the infimum is over all $\varepsilon$-DP mechanisms $\mathcal{M}$.*

*Proof of Theorem 7.* From Theorem 20, we have

$$\mathop{\mathbb{E}}_{\substack{D \sim P^n \\ P' \sim \mathcal{M}_{\varepsilon_1}^{\mathrm{Lap}}(D)}} [\|P' - P\|_1] \leq O\left(\sqrt{\frac{k}{n}} + \frac{k}{\varepsilon_1 n}\right).$$

Recall from Theorem 5 that there exists $\Phi : \mathcal{Y} \to \hat{\mathcal{Y}}$ such that $\mathcal{L}(\text{RR-on-Bins}_{\varepsilon}^{\Phi}; P) = \inf_{\mathcal{M}} \mathcal{L}(\mathcal{M}; P)$. Consider instead the $\text{RR-on-Bins}_{\varepsilon_2}^{\Phi}$ mechanism. We have

$$|\inf_{\mathcal{M}} \mathcal{L}(\mathcal{M}; P) - \mathcal{L}(\text{RR-on-Bins}_{\varepsilon_2}^{\Phi}; P)|$$

$$= |\mathcal{L}(\text{RR-on-Bins}_{\varepsilon}^{\Phi}; P) - \mathcal{L}(\text{RR-on-Bins}_{\varepsilon_2}^{\Phi}; P)|$$

$$\leq \sum_{y \in \mathcal{Y}} p_y \cdot B \cdot \left( \left| \frac{e^{\varepsilon}}{e^{\varepsilon} + |\hat{\mathcal{Y}}| - 1} - \frac{e^{\varepsilon_2}}{e^{\varepsilon_2} + |\hat{\mathcal{Y}}| - 1} \right| + \sum_{\hat{y} \in \hat{\mathcal{Y}} \setminus \{\Phi(y)\}} \left| \frac{1}{e^{\varepsilon} + |\hat{\mathcal{Y}}| - 1} - \frac{1}{e^{\varepsilon_2} + |\hat{\mathcal{Y}}| - 1} \right| \right)$$

$$= B \cdot \frac{2(|\hat{\mathcal{Y}}| - 1)(e^{\varepsilon} - e^{\varepsilon_2})}{(e^{\varepsilon} + |\hat{\mathcal{Y}}| - 1)(e^{\varepsilon_2} + |\hat{\mathcal{Y}}| - 1)}$$

$$\leq 2B \cdot (1 - e^{\varepsilon_2 - \varepsilon})$$

$$\leq 2B(\varepsilon - \varepsilon_2)$$

$$= 2B\varepsilon_1.$$

Recall that $\text{RR-on-Bins}_{\varepsilon_2}^{\Phi'}$ is an $\varepsilon_2$-DP optimal mechanism for prior $P'$. That is, we have

$$\mathcal{L}(\text{RR-on-Bins}_{\varepsilon_2}^{\Phi'}; P') \leq \mathcal{L}(\text{RR-on-Bins}_{\varepsilon_2}^{\Phi}; P').$$

Finally, we also have

$$|\mathcal{L}(\text{RR-on-Bins}_{\varepsilon_2}^{\Phi}; P) - \mathcal{L}(\text{RR-on-Bins}_{\varepsilon_2}^{\Phi}; P')| \leq B \cdot \|P' - P\|_1.$$

$$|\mathcal{L}(\text{RR-on-Bins}_{\varepsilon_2}^{\Phi'}; P) - \mathcal{L}(\text{RR-on-Bins}_{\varepsilon_2}^{\Phi'}; P')| \leq B \cdot \|P' - P\|_1.$$

Combining the above five inequalities, we arrive at

$$\mathbb{E}_{\substack{y_1, \ldots, y_n \sim P \\ P', \Phi', \hat{\mathcal{Y}}'}} [\mathcal{L}(\text{RR-on-Bins}_{\varepsilon_2}^{\Phi'}; P)] - \inf_{\mathcal{M}} \mathcal{L}(\mathcal{M}; P) \leq O\left( B \cdot \left( \varepsilon_1 + \sqrt{\frac{k}{n}} + \frac{k}{\varepsilon_1 n} \right) \right).$$

Setting $\varepsilon_1 = \sqrt{k/n}$ then yields the desired bound[7]. □

## D   DP MECHANISMS DEFINITIONS

In this section, we recall the definition of various DP notions that we use throughout the paper.

**Definition 21** (Global Sensitivity). Let $f$ be a function taking as input a dataset and returning as output a vector in $\mathbb{R}^d$. Then, the *global sensitivity* $\Delta(f)$ of $f$ is defined as the maximum, over all pairs $(X, X')$ of adjacent datasets, of $\|f(X) - f(X')\|_1$.

The (discrete) Laplace distribution with scale parameter $b > 0$ is denoted by $\text{DLap}(b)$. Its probability mass function is given by $p(y) \propto \exp(-|y|/b)$ for any $y \in \mathbb{Z}$.

**Definition 22** (Discrete Laplace Mechanism). Let $f$ be a function taking as input a dataset $X$ and returning as output a vector in $\mathbb{Z}^d$. The *discrete Laplace mechanism* applied to $f$ on input $X$ returns $f(X) + (Y_1, \ldots, Y_d)$ where each $Y_i$ is sampled i.i.d. from $\text{DLap}(\Delta(f)/\varepsilon)$. The output of the mechanism is $\varepsilon$-DP.

Next, recall that the (continuous) Laplace distribution $\text{Lap}(b)$ with scale parameter $b > 0$ has probability density function given by $h(y) \propto \exp(-|y|/b)$ for any $y \in \mathbb{R}$.

**Definition 23** (Continuous Laplace Mechanism, Dwork et al. (2006b)). Let $f$ be a function taking as input a dataset $X$ and returning as output a vector in $\mathbb{R}^d$. The *continuous Laplace mechanism* applied to $f$ on input $X$ returns $f(X) + (Y_1, \ldots, Y_d)$ where each $Y_i$ is sampled i.i.d. from $\text{Lap}(\Delta(f)/\varepsilon)$. The output of the mechanism is $\varepsilon$-DP.

---

[7]Note that this requires $n > k/\varepsilon^2$ for the setting of $\varepsilon_2 = \varepsilon - \varepsilon_1$ to be valid.

We next define the discrete and continuous versions of the staircase mechanism (Geng & Viswanath, 2014).

**Definition 24** (Discrete Staircase Distribution). Fix $\Delta \geq 2$. The *discrete staircase distribution* is parameterized by an integer $1 \leq r \leq \Delta$ and has probability mass function given by:

$$
p_r(i) = \begin{cases} a(r) & \text{for } 0 \leq i < r, \\ e^{-\varepsilon}a(r) & \text{for } r \leq i < \Delta \\ e^{-k\varepsilon}p_r(i - k\Delta) & \text{for } k\Delta \leq i < (k+1)\Delta \text{ and } k \in \mathbb{N} \\ p_r(-i) \text{ for } i < 0, \end{cases} \tag{15}
$$

where

$$
a(r) =:= \frac{1 - b}{2r + 2b(\Delta - r) - (1 - b)}.
$$

Let $f$ be a function taking as input a dataset $X$ and returning as output a scalar in $\mathbb{Z}$. The *discrete staircase mechanism* applied to $f$ on input $X$ returns $f(X) + Y$ where $Y$ is sampled from the discrete staircase distribution given in (15).

**Definition 25** (Continuous Staircase Distribution). The *continuous staircase distribution* is parameterized by $\gamma \in (0, 1)$ and has probability density function given by:

$$
h_\gamma(x) = \begin{cases} a(\gamma) & \text{for } x \in [0, \gamma\Delta) \\ e^{-\varepsilon}a(\gamma) & \text{for } x \in [\gamma\Delta, \Delta) \\ e^{-k\varepsilon}h_\gamma(x - k\Delta) & \text{for } x \in [k\Delta, (k+1)\Delta) \text{ and } k \in \mathbb{N} \\ h_\gamma(-x) \text{ for } x < 0, \end{cases} \tag{16}
$$

where

$$
a(\gamma) =:= \frac{1 - e^{-\varepsilon}}{2\Delta(\gamma + e^{-\varepsilon}(1 - \gamma))}.
$$

Let $f$ be a function taking as input a dataset $X$ and returning as output a scalar in $\mathbb{R}$. The *continuous staircase mechanism* applied to $f$ on input $X$ returns $f(X) + Y$ where $Y$ is sampled from the continuous staircase distribution given in (16).

**Definition 26** (Exponential Mechanism, McSherry & Talwar (2007)). Let $q(\cdot, \cdot)$ be a scoring function such that $q(X, r)$ is a real number equal to the score to be assigned to output $r$ when the input dataset is $X$. The *exponential mechanism* returns a sample from the distribution that puts mass $\propto \exp(\varepsilon q(X, r))$ on each possible output $r$. It is $(2\varepsilon\Delta(q))$-DP, where $\Delta(q)$ is defined as the maximum global sensitivity of $q(\cdot, r)$ over all possible values of $r$.

**Definition 27** (Randomized Response, Warner (1965)). Let $\varepsilon \geq 0$, and $q$ be a positive integer. The *randomized response* mechanism with parameters $\varepsilon$ and $K$ (denoted by $\mathsf{RR}_{\varepsilon,q}$) takes as input $y \in \{1, \ldots, K\}$ and returns a random sample $\tilde{y}$ drawn from the following probability distribution:

$$
\Pr[\tilde{y} = \hat{y}] = \begin{cases} \frac{e^{\varepsilon}}{e^{\varepsilon} + q - 1} & \text{for } \hat{y} = y \\ \frac{1}{e^{\varepsilon} + q - 1} & \text{otherwise.} \end{cases} \tag{17}
$$

The output of $\mathsf{RR}_{\varepsilon,q}$ is $\varepsilon$-DP.

# E  ADDITIONAL EXPERIMENT DETAILS

We evaluate the proposed RR-on-Bins mechanism on three datasets, and compare with the Laplace mechanism (Dwork et al., 2006b), the staircase mechanism (Geng & Viswanath, 2014) and the exponential mechanism (McSherry & Talwar, 2007). For real valued labels (the Criteo Sponsored Search Conversion dataset), we use the continuous Laplace mechanism (Definition 23) and the continuous staircase mechanism (Definition 25), and for integer valued labels (the US Census dataset and the App Ads Conversion Count dataset), we use the discrete Laplace mechanism (Definition 22) and the discrete staircase mechanism (Definition 24).

### E.1 Criteo Sponsored Search Conversion

The Criteo Sponsored Search Conversion dataset is publicly available from https://ailab.criteo.com/criteo-sponsored-search-conversion-log-dataset/. To predict the conversion value (SalesAmountInEuro), we use the following attributes as inputs:

- Numerical attributes: Time_delay_for_conversion, nb_clicks_1week, product_price.
- Categorical attributes: product_age_group, device_type, audience_id, product_gender, product_brand, product_category_1 ∼ product_category_7, product_country, product_id, product_title, partner_id, user_id.

All categorical features in this dataset have been hashed. We build a vocabulary for each feature by counting all the unique values. All the values with less than 5 occurrences are mapped to a single out-of-vocabulary item.

We randomly partition the dataset into 80%–20% train–test splits. For each evaluation configuration, we report the mean and std over 10 random runs. In each run, the dataset is also partitioned with a different random seed.

Our deep neural network consists of a feature extraction module and 3 fully connected layers. Specifically, each categorical attribute is mapped to a 8-dimensional feature vector using the pre-built vocabulary for each attribute. The mapped feature vectors are concatenated together with the numerical attributes to form a feature vector. Then 3 fully connected layers with the output dimension 128, 64, and 1 are used to map the feature vector to the output prediction. The ReLU activation is applied after each fully connected layer, except for the output layer.

We train the model by minimizing the mean squared error (MSE) with L2 regularization $10^{-4}$, using the RMSProp optimizer. We use learning rate 0.001 with cosine decay (Loshchilov & Hutter, 2017), batch size 8192, and train for 50 epochs. Those hyperparameters are chosen based on a setup with minor label noise (generated via Laplace mechanism for $\varepsilon = 6$), and then fixed throughout all runs.

For the RR-on-Bins mechanism, we use the recommended $\varepsilon_1 = \sqrt{|\mathcal{Y}|/n}$ in Theorem 7 to query a private prior, and with the remaining privacy budget, optimize the mean squared loss via dynamic programming. When running Algorithm 2, it would be quite expensive to use $\mathcal{Y}$ as the set of all unique (real valued) training labels. So we simply discretize the labels by rounding them down to integer values, and use $\mathcal{Y} = \{0, 1, \ldots, 400\}$. The integer labels are then mapped via Algorithm 1.

### E.2 US Census

The 1940 US Census can be downloaded from https://www.archives.gov/research/census/1940. We predict the time that the respondent worked during the previous year (the WKSWORK1 field, measured in number of weeks), based on the following input fields: the gender (SEX), the age (AGE), the marital status (MARST), the number of children ever born to each woman (CHBORN), the school attendance (SCHOOL), the employment status (EMPSTAT), the primary occupation (OCC), and the type of industry in which the person performed an occupation (IND). We use only 50,582,693 examples with non-zero WKSWORK1 field, and randomly partition the dataset into 80%/20% train/test splits. For each evaluation configuration, we report the mean and std over 10 random runs. In each run, the dataset is also partitioned with a different random seed.

Our deep neural network consists of a feature extraction module and 3 fully connected layers. The feature extraction module can be described via the following pseudocode, where the vocabulary size is chosen according to the value range of each field in the 1940 US Census documentation, and the embedding dimension for all categorical features are fixed at 8.

```
Features = Concat([
    Embedding{vocab_size=2}(SEX - 1),
    AGE / 30.0,
    Embedding{vocab_size=6}(MARST - 1),
    Embedding{vocab_size=100}(CHBORN),
    Embedding{vocab_size=2}(SCHOOL - 1),
    Embedding{vocab_size=4}(EMSTAT),
```

```
    Embedding{vocab_size=1000}(OCC),
    Embedding{vocab_size=1000}(IND),
])
```

The features vectors are mapped to the output via 3 fully connected layers with the output dimension 128, 64, and 1. The ReLU activation is applied after each fully connected layer, except for the output layer.

We train the model by minimizing the MSE with L2 regularization $10^{-4}$, using the RMSProp optimizer. We use learning rate 0.001 with cosine decay (Loshchilov & Hutter, 2017), batch size 8192, and train for 200 epochs. For the RR-on-Bins mechanism, we use the recommended $\varepsilon_1 = \sqrt{|\mathcal{Y}|/n}$ in Theorem 7 to query a private prior, and with the remaining privacy budget, optimize the mean squared loss via dynamic programming.

### E.3   App Ads Conversion Count Prediction

The app install ads prediction dataset is collected from a commercial mobile app store. The examples in this dataset are ad clicks, and each label counts post-click events (aka conversions) happening in the app after the user installs the app.

The neural network models used here is similar to the models used in other experiments: embedding layers are used to map categorical input attributes to dense feature vectors, and then the concatenated feature vectors are passed through several fully connected layers to generate the prediction. We use the Adam optimizer (Kingma & Ba, 2015) and the Poisson regression loss (Cameron & Trivedi, 2013) for training. For the RR-on-Bins mechanism, we use the recommended $\varepsilon_1 = \sqrt{|\mathcal{Y}|/n}$ in Theorem 7 to query a private prior, and with the remaining privacy budget, optimize the mean squared error via dynamic programming. Following the convention in event count prediction in ads prediction problems, we train the loss with the Poisson log loss. We report the relative Poisson log loss on the test set with respect to the non-private baseline.

## F   Label Clipping

The Laplace mechanism and staircase mechanism we compared in this paper could both push the randomized labels out of the original label value ranges. This issue is especially severe for small $\varepsilon$'s, as the range of randomized labels could be orders of magnitude wider than the original label value range. Since the original label value range is known (required for deciding the noise scale of each DP mechanism), we could post process the randomized labels by clipping to this range.

We compare the effects of clipping for both Laplace mechanism and staircase mechanism on the Criteo dataset in Figure 6. In both case, this simple mitigation helps reduce the error significantly, especially for smaller epsilons. So in all the experiments of this paper, we always apply clipping to the randomized labels.

Figure 6 also shows the exponential mechanism (McSherry & Talwar, 2007), which is equivalent to restricting the Laplace noises to the values that would not push the randomized labels out of the original value range. In our case, we implement it via rejection sampling on the standard Laplace mechanism. This algorithm avoid the boundary artifacts of clipping, but to guarantee the same level of privacy, the noise needs to scale with $2/\varepsilon$, instead of $1/\varepsilon$ as in the vanilla Laplace mechanism. As a result, while it could be helpful in the low epsilon regime, in moderate to high $\varepsilon$ regime, it is noticeably worse than all other methods, including the vanilla Laplace mechanism due to the $2\times$ noise scaling with $1/\varepsilon$.

## G   Comparison with `RRWithPrior`

In this paper, we extended the formulation of Ghazi et al. (2021a) from classification to regression losses. Here we provide a brief comparison between our algorithm (RR-on-Bins) and the `RRWithPrior` algorithm from Ghazi et al. (2021a) on the US Census dataset. Instead of using multi-stage training, we fit `RRWithPrior` in our feature-oblivious label DP framework, and use the same privately queried global prior as the input to both RR-on-Bins and `RRWithPrior`. For `RRWithPrior`,

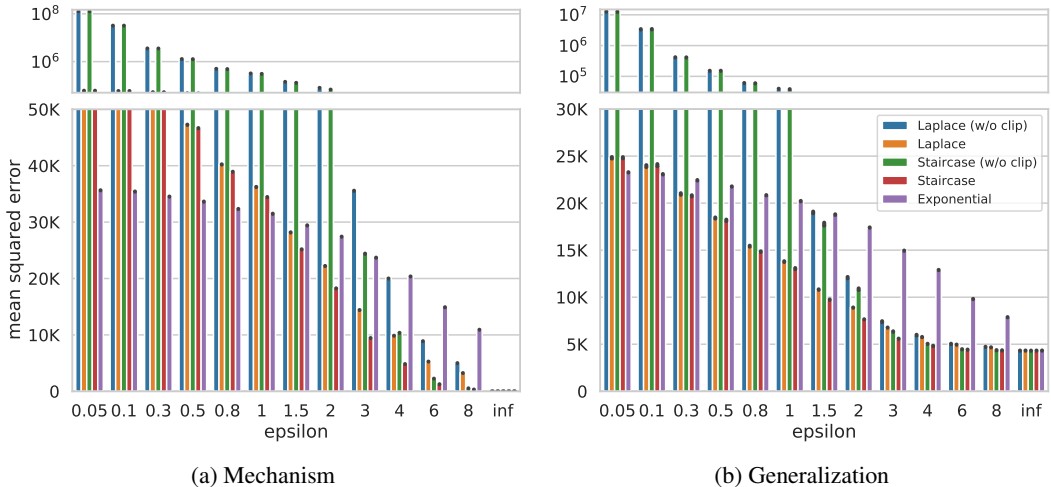

|  | (a) Mechanism | (b) Generalization |
|---|---|---|

Figure 6: MSE on the Criteo dataset, with or without postprocess clipping. (a) measures the error introduced by each DP randomization mechanism on the *training* labels. (b) measures the *test* error of the models trained on the corresponding private labels.

| Privacy Budget | MSE (Mechanism) | | MSE (Generalization) | |
|---|---|---|---|---|
| | RRWithPrior | RR-on-Bins | RRWithPrior | RR-on-Bins |
| 0.5 | 274.11 | 179.88 | 273.97 | 183.28 |
| 1.0 | 274.11 | 159.49 | 273.97 | 172.64 |
| 2.0 | 274.11 | 107.89 | 273.97 | 152.03 |
| 4.0 | 138.00 | 36.43 | 145.89 | 136.59 |
| 8.0 | 11.58 | 2.54 | 134.26 | 134.35 |
| $\infty$ | 0.00 | 0.00 | 134.27 | 134.27 |

Table 2: MSE on the US Census dataset, comparing RR-on-Bins with RRWithPrior. The first block (Mechanism) measures the error introduced by the DP randomization mechanisms on the training labels. The second block (Generalization) measures the test error of the models trained on the corresponding private labels. Note RRWithPrior was designed for classification loss (Ghazi et al., 2021a), here we just ignore the numerical similarity metric and treat each integer label as a separate category.

we simply treat each of the integer label values in US Census as an independent category during the randomization stage. Once the private labels are obtained, the training stage is identical for both algorithms. The results are shown in Table 2. Note the results are for reference purposes, as this is not a fair comparison. Because RRWithPrior was designed for classification problems (Ghazi et al., 2021a), which ignore the similarity metrics between different labels, while RR-on-Bins takes that into account.

