# OpenReview forum: "Regression with Label Differential Privacy"
_ICLR.cc/2023/Conference — ICLR 2023 poster_

### Official Review · Reviewer_Qh4U · 2022-10-22

**Confidence:** 5
**Correctness:** 3
**Technical Novelty And Significance:** 3
**Empirical Novelty And Significance:** 3
**Recommendation:** 6

**Clarity, Quality, Novelty And Reproducibility:**

The authors try to extend the method in paper "Deep Learning with Label Differential Privacy" for classification to regression. The authors try to use step function to represent the tilde  Y space. It seems a good paper if the derivation is correct.



**Strength And Weaknesses:**

Strength: The authors try to extend the method and results of labeled DP solving classification problem to regression. The problem is good.
Weaknesses: They map the response (maybe in a continuous region (interval)) to finite possible values.

**Summary Of The Paper:**

The authors study the task of training regression models with the guarantee of label differential privacy (DP). Based on a global prior distribution on label values, which could be obtained privately, they derive a label DP randomization mechanism that is optimal under a given regression loss function.

**Summary Of The Review:**

It is an interesting problem.

There are some typos, authors should read the draft once more before submission.

---

> ### Author Response · Authors · 2022-11-12
> **Regression vs classification**
>
> We would like to point out that the main difference between _regression_ and _classification_ is that for regression tasks, the loss function (e.g., the mean-squared loss or the Poisson loss) assigns smaller loss values to closer numbers. By contrast, classification losses treat every incorrect class equally.
> Also note that Poisson regression (which is well-studied in ML and statistics) is about regressing against a non-negative _integer count_, which is not an uncountable set.
>
> Finally, we have revised our submission to include in the Appendix (Theorem 15 in Appendix A.1), an extension of Theorem 5, which shows that RR-on-Bins with a finite output set $\hat{Y}$  is optimal even when the set of possible labels $Y$ is a continuous (infinite) bounded interval.

---

> > ### Comment · Reviewer_Qh4U · 2022-11-16
> > **loss function**
> >
> > It may not be proper to distinguish regression and classification. It is also okay to use mean-squared loss when choosing y as the K-dimension probability vector for classification task with K categories.
> >
> > Thanks for your reply. We indeed made a mistake that we thought your Y is also finite. Good luck.

---

> > > ### Author Response · Authors · 2022-11-17
> > > **Further clarification on regression vs classification**
> > >
> > > We thank the reviewer for raising the score and additional comments!
> > >
> > > We would like to further clarify regarding regression. Indeed we can use mean-squared loss on one-hot encoded label vectors with classification problems. But that still does not encode a similarity metric between different classes like what we have in the regression case. For example, if we have three classes: cat, dog, duck. The MSE loss between cat and dog is sum([1, 0, 0] - [0, 1, 0])^2 = 4, which is the same as the loss between cat and duck. However, in the case of regression, some classes are more similar to each other than others. For example, if we have three (say integer) labels 1, 2, 3, then the square loss between label-1 and label-3 is 4, but the square loss between label-1 and label-2 is only 1. Assumption 4 in our paper covers many of the loss functions induced by these similarity metrics. Hope this clarifies the difference.

---

### Official Review · Reviewer_hzx9 · 2022-10-25

**Confidence:** 2
**Correctness:** 4
**Technical Novelty And Significance:** 2
**Empirical Novelty And Significance:** 2
**Recommendation:** 6

**Clarity, Quality, Novelty And Reproducibility:**

This paper is well written. I enjoyed reading this paper. Compared to Ghazi et al. (2021a), the proposed method is not technically novel.


**Strength And Weaknesses:**

Strengths: 1. This paper is clearly written. Also, it provides extensive experiments on real data.

Limitations: 1. In the practical setting, the label party is typically able to see the public feature data.
2. In practice,  the overall generalization error is usually more concerning. I was wondering if there is any way to also consider the second term in Eq. (1) like in Ghazi et al. (2021a)?


**Summary Of The Paper:**

This paper provides an optimal label randomization mechanism that ensures local differential privacy given the prior distribution of labels. For a known loss function, the proposed mechanism is optimal. If the prior is unknown, this paper proposes to use the private histogram to estimate the prior. In this case, the gap between the proposed method and the optimal mechanism is proved to be O(\sqrt{|Y|/n}), where $|Y|$ is the cardinality of the label domain. Compared to prior work Ghazi et al. (2021a), the proposed mechanism works for general regression losses. This paper also empirically compares the label loss of the proposed method with some baselines on real data.

**Summary Of The Review:**

This paper provides an optimal mechanism for privatizing the labels that minimize the loss between noisy labels and true labels. The proposed method is a strict generalization of 0/1 loss appeared in Ghazi et al. (2021a). Theoretically, this paper shows the optimality of the proposed mechanism. Empirically, the authors demonstrate the effectiveness through extensive experiments.

---

> ### Author Response · Authors · 2022-11-12
> **Linear program**
>
> 1. Our algorithm is more general and could be applied both when the label party does and does not have access to the input features.
> 2. We note that the second term in (1) can be thought of as an optimization problem on the distribution of $(x, \hat{y})$ instead of the original distribution $(x, y)$. Therefore, if the training algorithm has a good generalization bound with regards to an arbitrary distribution, then the bound can also be applied to the second term in the RHS of (1).
>
> We would like to point out the actual linear program (LP), its solution structure, and the resulting algorithm are entirely different from that of Ghazi et a (or for that matter, LPs that have been used in earlier works, e.g., [Ghosh et al., STOC 2009](https://dl.acm.org/doi/10.1145/1536414.1536464), [Kairouz et al., NeurIPS 2014](https://proceedings.neurips.cc/paper/2014/hash/86df7dcfd896fcaf2674f757a2463eba-Abstract.html)). Due to the properties of the solution, we are able to design a simple dynamic programming solution for our problem; such a step was not needed in Ghazi et al. In fact, characterizing the solution structure from the LP (Theorem 5) is one of the main novelties of our work (Appendix A contains a complete proof of this characterization); the LP studied in Ghazi et al. turned out to have a far simpler solution because of their focus on the classification loss.

---

### Official Review · Reviewer_deEc · 2022-10-25

**Confidence:** 4
**Clarity, Quality, Novelty And Reproducibility:** The clarity is good and the method is…
**Correctness:** 4
**Technical Novelty And Significance:** 4
**Empirical Novelty And Significance:** 4
**Recommendation:** 8

**Strength And Weaknesses:**

**Strength**
1. The writing of this paper is very clear.
2. This paper proposes a novel label-DP algorithm and proves its optimality w.r.t. the expected loss between the original label and the flipped labels.
3. The experiments are conducted on the real world datasets and the results are convincible.

**Weaknesses**
1. As stated, the proposed label-DP mechanism is optimal w.r.t. the first term in the RHS of Equation (1). However, the mechanism will also influence the minimum of the second term, which is the bayes optimal error to predict $\hat{y}$ from $x$. Think of this simplified setting: when $y$ is deterministic given $x$, the bayes optimal error is equivalent to the variance of $\hat{y}$ condition on $y$. By considering this additional variance, it is potential to come up with a better label-DP mechanism w.r.t. the entire RHS of Equation (1).
2. Is it possible that the number of optimal bins is smaller than the number of different labels?

**Summary Of The Paper:**

This paper studies the regression task with label differential privacy. It proposes a label flipping mechanism with the guarantee of $(\varepsilon, \delta)$-label DP and further proves that it is the optimal label-DP mechanism which achieves the minimum expected loss between the original label and the flipped label. Finally it evaluates the proposed label DP mechanism and classical Laplace Mechanism on three different datasets. The result shows the efficacy of the proposed mechanism.

**Summary Of The Review:**

Given that the algorithm is novel and proved to be optimal in a general setting, I give the score of 8.

---

> ### Author Response · Authors · 2022-11-12
> **Eq (1) and the number of optimal bins**
>
> 1. We thank the reviewer for the suggestion regarding the Bayes optimal error in the simplified setting.  We agree that this is an interesting research direction, and that methodologies other than ours (which uses the triangle inequality in (1)) could be explored in future work ; we will add a corresponding discussion in the revision.
>
> 2. Yes, the number of optimal bins could be smaller than the number of different labels. Interestingly, we empirically observe that the number of bins decreases with decreasing $\epsilon$. However, we do not yet have a formal proof that this monotonicity always holds.

---

### Official Review · Reviewer_dKhp · 2022-10-27

**Confidence:** 3
**Clarity, Quality, Novelty And Reproducibility:** paper is clearly written and is easy …
**Correctness:** 3
**Technical Novelty And Significance:** 3
**Empirical Novelty And Significance:** 3
**Recommendation:** 8

**Strength And Weaknesses:**

Proposed method is interesting and of practical use where one might want label differential privacy. Empirical evaluation shows that the proposed method performs better than other common DP methods such as Laplace, Exponential, and staircase.

Paper is clearly written and is easy to follow. Assuming the proofs go through (I did not check them in detail), I think the paper will make positive contribution to the DP community. My only question is that there should be some discussion on why it is not compared against other label-DP methods such as [1].

[1] Ghazi, B., Golowich, N., Kumar, R., Manurangsi, P., & Zhang, C. (2021). Deep learning with label differential privacy. Advances in Neural Information Processing Systems, 34, 27131-27145.

**Summary Of The Paper:**

Paper proposes regression with label differential privacy and introduces randomized response on bins as a new mechanism to induce differential privacy on labels.

**Summary Of The Review:**

paper is easy to follow and has the potential for a net positive impact.

---

> ### Author Response · Authors · 2022-11-12
> **Comparison against [1]**
>
> Thank you for raising this point.  We adapted the algorithm from [1] to our framework and evaluated it on the US Census data by treating each integer-valued label as a categorical label.  The results are shown below.
>
>   ε | RR-on-Bins | Alg in [1]
> ---:|:---:|:---:
>  0.5  |    183.28     |   273.97
>  1.0  |    172.64     |   273.97
>  2.0  |    152.03     |   273.97
>  4.0  |    136.59     |   145.89
>  8.0  |    134.26     |   134.35
>   ∞   |    134.27     |   134.27
>
> As can be seen, RR-on-Bins does much better than the algorithm in [1].  However, we omitted these results in our submission since it is not a fair comparison: the algorithm in [1] is designed for classification and hence treats every incorrect class equally whereas RR-on-Bins is designed for regression loss, which scales with the distance between the predicted value and the label value. If you feel it would add value, we are happy to include these results in the revision.

---

> > ### Comment · Reviewer_dKhp · 2022-11-15
> > **comparison against [1]**
> >
> > Thank you for the response and for the additional experiment. I agree that the comparison is not fair, but I do think that it helps readers understand the contribution of this paper better. I think it would be nice to include it in the appendix with a discussion as to why it is not a fair comparison. As the authors have answered my question, I am increasing my score.

---

> > > ### Author Response · Authors · 2022-11-18
> > > **Including the comparison in the paper**
> > >
> > > Thanks for raising the score! We have added a new section (Appendix G) to include these results in the revised paper.

---

### Decision · Program_Chairs · 2023-01-20

**Decision:**

Accept: poster

**Justification For Why Not Higher Score:**

There are better papers in my pile that are more deserving of spotlight.

**Justification For Why Not Lower Score:**

This paper is clearly not in the reject pile.

**Metareview: Summary, Strengths And Weaknesses:**

This paper considers regression tasks taking into account label differential privacy.  It proposes a label flipping mechanism with the guarantee of $(\varepsilon,\delta)$-label DP and further proves that it is the optimal label-DP mechanism which achieves the minimum expected loss between the original label and the flipped label. Finally it evaluates the proposed label DP mechanism and classical Laplace Mechanism on three different datasets.

Generally, the reviewers found the contributions in the paper to be novel and significant. The experimental evaluations are also sufficient. During the discussion phase, the reviewers became more convinced about the contributions, and raised their scores.

Please incorporate the additional experimental results comparing your method to [1] in the final version of the paper.

[1] Ghazi, B., Golowich, N., Kumar, R., Manurangsi, P., & Zhang, C. (2021). Deep learning with label differential privacy. Advances in Neural Information Processing Systems, 34, 27131-27145.

**Note From Pc:**

if the above contains the word "oral" or "spotlight" please see: "oral" presentation means -> notable-top-5% and "spotlight" means -> notable-top-25%. As stated in our emails, we are disassociating presentation type from AC recommendations

**Summary Of Ac-Reviewer Meeting:**

NIL